# The pupillary respiratory-phase response: pupil size is smallest around inhalation onset and largest during exhalation

Martin Schaefer[1] (iD), Sebastiaan Mathôt[2] (iD), Mikael Lundqvist[1,3] (iD), Johan N. Lundström[1,4,5] (iD) and Artin Arshamian[1] (iD)

[1] *Department of Clinical Neuroscience, Karolinska Institutet, Stockholm, Sweden*
[2] *Department of Psychology, University of Groningen, Groningen, The Netherlands*
[3] *Department of Brain and Cognitive Sciences, Massachusetts Institute of Technology, Cambridge, MA, USA*
[4] *Monell Chemical Senses Center, Philadelphia, PA, USA*
[5] *Department of Otorhinolaryngology, Karolinska University Hospital, Stockholm, Sweden*

Handling Editors: Harold Schultz & Daniel Zoccal

The peer review history is available in the Supporting Information section of this article (https://doi.org/10.1113/JP287205#support-information-section).

**Abstract figure legend** Across five experiments including 203 participants and a wide range of experimental conditions we demonstrate that pupil size fluctuates in a systematic manner over the course of the breathing cycle. Specifically, we show that pupil size is largest during exhalation and smallest around inhalation onset. We term this effect the pupillary respiratory-phase response.

**Martin Schaefer** is a recent PhD graduate from the Department of Clinical Neuroscience at the Karolinska Institute. His doctoral research focused on respiratory influences on pupil size and visual recognition memory, and he currently investigates breathing's effects on pupil dynamics and visual perception. His broader research interests encompass interactions between physiology, brain activity, behaviour and perception. As a member of the Embodied Perception and Cognition Group, he also explores the neuroscience and psychology behind the sense of smell.

This article was first published as a preprint. Schaefer M, Mathôt S, Lundqvist M, Lundström JN, Arshamian A. 2024. The respiratory-pupillary phase effect: Pupils size is smallest around inhalation onset and largest during exhalation. bioRxiv. https://doi.org/10.1101/2024.06.27.599713

**Abstract** Respiration shapes brain activity and synchronizes sensory and exploratory motor actions, with some evidence suggesting that it also affects pupil size. However, evidence for a coupling between respiration and pupil size remains scarce and inconclusive, hindered by small sample sizes and limited controls. Given the importance of pupil size in visual perception and as a reflection of brain state, understanding its relationship with respiration is essential. In five experiments using a pre-registered protocol, we systematically investigated how respiratory phase affects pupil size across different conditions. In Experiment 1 ($n = 50$), we examined nasal and oral breathing at rest under dim lighting with nearby fixation points, then replicated these results under identical conditions in Experiment 2 ($n = 53$). Experiment 3 ($n = 112$) extended this to active visual tasks, while Experiment 4 ($n = 57$) extended this to controlled breathing at different paces under ambient lighting with distant fixation. Finally, in Experiment 5 ($n = 34$), individuals with isolated congenital anosmia (born without olfactory bulbs) were used as a lesion-type model during visual–auditory tasks to assess whether the respiratory–pupil link depends on olfactory bulb-driven oscillations.

Across all conditions – free and controlled breathing; different tasks, lighting and fixation distances; and with and without olfactory bulbs – we consistently found that pupil size is smallest around inhalation onset and largest during exhalation. We term this effect the pupillary respiratory-phase response, the fourth known mechanism influencing pupil size, alongside the pupillary light, near fixation and psychosensory responses.

(Received 2 July 2024; accepted after revision 31 January 2025; first published online 21 February 2025)

**Corresponding authors** M. Schaefer and A. Arshamian: Department of Clinical Neuroscience, Karolinska Institutet, Stockholm, Sweden.    Email: martin.schaefer@ki.se and artin.arshamian@ki.se

## Key points

- The influence of respiration on pupil size dynamics has long been debated.
- In this study, we systematically investigated how pupil size changes across the breathing cycle through a series of five experiments, while varying tasks, lighting, fixation distance and brain region involvement.
- We show that pupil size is smallest around inhalation onset and largest during exhalation, with pupil dilatation occurring through most of inhalation and the early phase of exhalation, and pupil constriction occurring primarily during the latter part of exhalation.
- This pattern was consistent across all experimental conditions, demonstrating that it is robust and likely controlled by brainstem circuits.
- We term this effect the pupillary respiratory-phase response, the fourth known mechanism influencing pupil size.

## Introduction

Respiration plays a crucial role in regulating various brain and behavioural functions beyond oxygenation. It does so by generating neural oscillations that propagate globally across the brain, which in turn affect behaviour (Arshamian et al., 2018; Heck et al., 2017; Ito et al., 2014; Kleinfeld et al., 2014; Kluger & Gross, 2020, 2021; Perl et al., 2019; Tort et al., 2018; Zelano et al., 2016). These oscillations are driven by the brainstem's pre-Bötzinger complex (preBötzC; del Negro et al., 2018) and (for nasal respiration only) the olfactory bulb, thus creating respiratory-coupled oscillations that influence perception and cognition (Arshamian et al., 2018; del Negro et al., 2018; Fontanini & Bower, 2006; Heck et al., 2017; Karalis & Sirota, 2022; Kay et al., 2009; Kluger et al., 2021; Kocsis et al., 2018; Perl et al., 2019; Schaefer et al., 2024; Tort et al., 2018; Zelano et al., 2016). In animal models, respiratory-coupled oscillations, particularly those originating from brainstem circuits, act as a 'master clock', synchronizing sensory and exploratory motor actions such as whisking, sniffing, licking, bobbing and head moving (Kleinfeld et al., 2014, 2023).

Similarly, for a long time it has been suggested that respiration shapes pupil size, leading to pupil dilatation during inhalation and pupil constriction

during exhalation (Ashhad et al., 2022; Borgdorff, 1975). However, a recent systematic review by Schaefer et al. (2022) highlighted that the empirical support that respiratory phase shapes pupil size is scarce, contradictory and inconclusive (e.g. Borgdorff, 1975; Nakamura et al., 2019; Schaefer et al., 2022). The lack of evidence primarily stems from the fact that only a handful of studies have directly tested the effect of respiration on pupil size, and these studies have generally suffered from methodological issues, such as small sample sizes, insufficient or absent statistical analyses, inadequate physiological measurements and failure to control for eye blinks (Schaefer et al., 2022). This is unfortunate because, while respiration influences overall brain function and behaviour, pupil size shapes visual perception and its fluctuations provide a dynamic readout of brain and behavioural states (Joshi & Gold, 2020; Mathôt, 2018; Schwalm & Rosales Jubal, 2017). Given this, understanding the interaction between breathing and pupil size is important.

Therefore, we systematically investigated the relationship between respiration and pupil size in a series of five experiments, covering various respiration types and a wide range of conditions known to engage specific brain regions or affect pupil size.

## Methods

### Ethical approval

Ethical approval was obtained from the national Swedish Ethical Review Authority (Dnr. 2020-00972 for Experiments 1–4, and Dnr. 2020–06533 for Experiment 5) and all participants signed informed written consent prior to participation. The study conformed to the standards set by the *Declaration of Helsinki*, except for the lack of registration in a database before recruitment of the first participant (clause 35).

### Participants

For Experiments 1–3, we recruited a total of 112 participants (70 female) ranging in age from 18 to 42 years (mean age 28.0 years). Of these, 50 participants (35 female) ranging in age from 18 to 42 years, with a mean age of 27.9 years, were part of Experiment 1. Another 53 participants (30 female) ranging in age from 19 to 42 years with a mean age of 28.5 years, were part of Experiment 2. All 112 participants participated in Experiment 3. Three participants did not return for their second testing session (affecting Experiments 1 and 3).

For Experiment 4 we recruited 57 participants (35 female) ranging in age from 18 to 46 years with a mean age of 28.5 years. Two participants (one female) had to be excluded from the experiment due failure of the eye tracker to record data, or due to reduced vision of the participant.

For Experiment 5, a subset of participants who were recruited for previous magnetic resonance imaging studies were analysed for the current study (Peter et al., 2023; Thunell et al., 2024). The subset of 34 participants was selected for this study based on their valid eye-tracking data (eye tracking was measured for more participants, but in many cases the measurement was not successful due to obscured eyes by the head coil or a change in the participant's position after the calibration of the eye tracker). This subset consisted of 21 participants (12 female) with isolated congenital anosmia, an exceedingly rare disorder, ranging in age from 21 to 50 years old, with a mean age of 34.1 years and 13 control participants (nine female) with a normal sense of smell, ranging in age from 18 to 48 years, with a mean age of 32.9 years. Participants with congenital anosmia all lacked bilateral olfactory bulbs and control participants all had clearly identifiable olfactory bulbs (assessed from obtained structural images by J.N.L.; see Peter et al., 2023 and Thunell et al., 2024 for more information).

All participants reported being healthy and able to breathe freely and independently through both their nose and mouth. Participants were instructed to refrain from ingesting caffeine on the day of the experiment as this is known to affect pupil size dynamics (Abokyi et al., 2017; Redondo et al., 2020). Except for participants in Experiment 5, they received compensation for their participation in the form of three movie ticket vouchers or three gift vouchers with a combined value of 300 SEK.

### Experimental setup

**Experiments 1 and 2 (replication): at rest.** Participants underwent two 5 min sessions at rest during which they looked at a grey screen with a black fixation cross. In one session, they only breathed through their nose, and in the other, only through their mouth. The order of which breathing route came first was balanced across participants, and the sessions were conducted on separate days.

The participants were seated with their heads fixed in a chin rest in a dimly lit room, without any variation in light intensity through the experiments, at a distance of 61 cm from the screen. The participants were instructed to breathe normally and to keep their eyes open (blinking allowed).

**Experiment 3: visual task.** We conducted a three-alternative forced choice visual detection task (modelled after Kluger et al., 2021) in the same luminance condition as during Experiments 1 and 2. During the

task, participants fixated on a cross (0.4° in diameter) in the centre of the screen presented against a grey background. Each trial consisted of a fixation period (jittered between 1200 and 3500 ms), followed by a brief (50 ms) presentation of a small (0.3° in diameter) Gabor patch in a marked circular area (3.5° in diameter) 10° to the left or the right side of the fixation cross, or no Gabor patch was presented (catch trials). After a delay of 500 ms, a question mark in the centre of the screen prompted participants to press an arrow key to indicate whether they had seen a Gabor patch, and if so on which side of the screen. After the participants pressed one of the arrow keys, the next trial started. For each trial, target contrast was adapted by a QUEST staircase (Watson & Pelli, 1983) aimed at individual hit rates of about 60%. The task consisted of 762 trials with an equal number of left targets, right targets and catch trials. The first 12 trials of the task were practice trials, and after every 30 trials, the participants got a short *ad libitum* break. The behavioural results of this task are not part of this study and will be presented elsewhere. The pupil size recordings during the breaks were not analysed as participants were free to remove their head from the chin rest and move around (while staying seated).

**Experiment 4: controlled breathing.** Participants had to undergo a series of three breathing tasks lasting 5 min each. The tasks consisted of a sequence of normal breathing at rest, controlled slow-paced breathing, and controlled fast-paced breathing, where the order of slow- and fast-paced breathing was alternated between participants. Participants were seated in a brightly lit room and placed their head on a chin rest. During all breathing tasks, participants fixed their gaze at a red fixation cross taped to the door behind them, viewed via a mirror in front of them. The distance from the chin rest to the mirror was 137 cm, and the distance from the mirror to the fixation cross was 207 cm. During the slow- and fast-paced breathing blocks, participants regulated their breathing by aligning their respiratory rate to alternating tones played at eight breaths/min for slow breathing and 16 breaths/min for fast breathing. Tones were presented via over-ear headphones worn during the paced breathing blocks. All breathing blocks were performed using nose-breathing only.

**Experiment 5: with and without olfactory bulbs.** Participants underwent olfactory assessment using the full Sniffin' Sticks testing procedure, which consisted of olfactory detection threshold, odour quality discrimination and cued odour identification ability (Hummel et al., 2007). Following the olfactory assessment, participants underwent functional magnetic resonance imaging, using a 3T Magnetom Prisma Scanner (Siemens) with a 20-channel head coil. The functional imaging consisted of four runs lasting 10 min and 15 s each. Each run contained eight blocks, and within each block six auditory or visual stimuli were presented for a duration of 3 s each. The stimulus set comprised grayscale pictures and sounds of 48 easily identifiable objects, chosen to evoke various degrees of odour association. The participants had been presented with all of the stimuli before the scanning and were tasked with identifying the stimuli during the scan, but otherwise had no task besides attending to the stimuli. Between stimuli presentation the participants were asked to look at a fixation cross at the centre of the screen. Auditory stimuli were presented via binaural headphones (NordicNeuroLab AS, Bergen, Norway), at a loud volume that was adjusted when needed to be tolerable to the participant. Visual stimuli were presented on a 40" 4K ultra-HD LCD display (NordicNeuroLab AS, Bergen, Norway) placed behind the scanner bore and viewed via a head coil mirror. Viewing distance was approximately 180 cm, and the pictures spanned 4.5° of visual angle. For the duration of the scanning procedure, participants only breathed through their nose. For a more detailed description of the procedure, please see Thunell et al. (2024).

## Physiological measures

Breathing was measured in two ways for Experiments 1–3. We used a non-restrictive breathing mask attached to a breathing cannula, which in turn was connected to a spirometer pod (ML311, ADInstruments, Colorado, US) to obtain a direct measure of airflow (0202-1199, Tiga-Med, Germany). Additionally, we used a temperature probe (MLT415/D, ADInstruments, Colorado, US) attached to the inside of the breathing mask and recorded breathing by measuring changes in air temperature over time ($\Delta$T°C). Using two breathing measures gave us a backup in case one of the measures malfunctioned and allowed us to use whichever measure contained less noise. The breathing route not assessed was covered topically with surgical tape to prevent participants from accidentally using the wrong breathing route. In Experiments 4 and 5, breathing was measured only using a dual port nasal cannula (Salter Labs), without the temperature measurements. All breathing signals were amplified, digitized at 1000 Hz (LabChart 7.0, ADInstruments, Colorado, US), and low pass-filtered at 5 Hz (PowerLab 16/35, ADInstruments, Colorado, US). In Experiments 1–4, pupil size was measured with a Gazepoint eye tracker with a sampling rate of 60 Hz and recorded with the Gazepoint software. Importantly, in Experiment 4, we used an updated version of the Gazepoint software which allowed for a more accurate recording of absolute pupil size in

millimetres. In Experiment 5, a TRACKPixx3 2 kHz binocular distance eye tracker was used to measure pupil size (VPixx Technologies Inc., Saint-Bruno, QC Canada). Heart rate was measured in Experiments 1–4; using a finger pulse electrode, and a three-leads ECG setup (LabChart 7.0, ADInstruments, Colorado, US). Heart rate data were collected as a control measure for the behavioural task of Experiment 3, which will not be included here.

## Pre-processing of data

**Pre-registration.** We conducted an exploratory analysis for Experiment 1. Based on those results, we pre-registered the pre-processing and analysis steps for Experiment 2 on the Open Science Framework (https://osf.io/xwjqe). These pre-registered steps were subsequently applied to Experiments 3–5 as well and are detailed below.

**Pupil data.** The pre-processing of the pupil size data was done in several stages, separately for each eye, and mainly followed the suggestions of Mathôt & Vilotijević (2022).

First, any recording marked by the eye tracker as valid but shorter than 250 ms in duration was marked as invalid to exclude data points with unreliable validity. In addition, valid data shorter than 500 ms which were surrounded by invalid data of more than 2 s, were also marked as invalid.

Second, any recording marked as invalid by the eye tracker for more than 500 ms was removed to exclude invalid data that were unlikely to be blinks.

Third, any recording marked as invalid by the eye tracker, or our above criteria was interpolated using the *fillmissing.m* Matlab function employing a piecewise cubic spline interpolation method. For each invalid data stretch, a buffer of two frames before and after was marked as invalid as well, and then the complete stretch was interpolated using the three valid samples preceding and following the buffer frames. If the range of the interpolated data points exceeded the range of the valid data points used for the interpolation by more than 30% (which might indicate that spikes had been introduced by the interpolation), the interpolation was redone using three and then four buffer frames around the stretch of invalid data points. If the range of the interpolated data points still exceeded the range of the valid data points used for the interpolation by more than 30%, a linear interpolation using the two buffer frames and three valid samples around the stretch of invalid data points was used instead.

Fourth, the data were checked for rapid changes in pupil size that could indicate blinks or invalid interpolations. The stretches of data detected with this method were then interpolated in the same manner as described above. However, the cubic spline interpolation was only deemed valid if its range did not exceed the range of the valid

samples used for the interpolation by 20%. After finishing the interpolations, any valid stretches of data shorter than 250 ms in duration were marked as invalid to exclude data points with unreliable validity. In addition, valid data stretches shorter than 500 ms that were surrounded by invalid data stretches of more than 2 s, and valid data stretches shorter than 1000 ms that were surrounded by invalid data stretches of more than 5 s were also marked as invalid.

Additionally, data values smaller or larger than the mean pupil size $\pm 3 \times$ the standard deviation of the pupil size values were marked invalid.

All interpolated pupil data were visually inspected to make sure the data looked reasonable, and participants with more than 30% invalid pupil data for both eyes after interpolation were excluded from further analysis. For Experiment 5, we rejected individual runs with more than 30% invalid pupil data for both eyes, rather than rejecting at the participant level.

After rejecting recording sessions with too much invalid pupil data, we were left with 38 complete participants (41 nose-breathing sessions and 44 mouth-breathing sessions) for Experiment 1, and 44 complete participants (48 nose-breathing sessions and 47 mouth-breathing sessions) for Experiment 2. For the visual detection task, we had 98 complete participants (103 nose-breathing sessions and 102 mouth-breathing sessions). For Experiment 4, we were left with 51 complete participants (53 normal-, 53 slow-, and 55 fast-breathing blocks). For Experiment 5, we rejected 42 out of 136 runs due to too little valid eye-tracking data, resulting in 25 runs from 13 participants with olfactory bulbs, and 69 runs from 21 participants without olfactory bulbs.

In line with our pre-registration, we also assessed changes in absolute pupil size over time in Experiments 1 and 2. To get the absolute pupil size values, we repeated the pupil interpolation in the same manner as for the pupil size values in pixels. However, the pupil size recordings in absolute values (mm) were less reliable than the recordings of the pupil size in pixels and we were therefore left with fewer valid recordings. After rejecting recording sessions with too much invalid pupil data, we were left with 30 complete participants (35 nose-breathing sessions and 39 mouth-breathing sessions) for Experiment 1, and 41 complete participants (47 nose-breathing sessions and 43 mouth-breathing sessions) for Experiment 2.

To increase power and reliability, we used the average of the left and right eye for each analysis where we looked at pupil size. Furthermore, we normalized all relative pupil size data (pixels) by *z*-scoring with mean zero and standard deviation of one, at the participant level to enable direct comparisons across participants.

For Experiment 5, the pupil size data were down-sampled to 100 Hz before the pre-processing to reduce the size of the data.

**Breathing data.** Given that the airflow data exhibited a higher temporal sensitivity than the thermopod data, we used the airflow data to determine the breathing phase for each participant. To analyse the respiratory patterns of each participant, we employed the BreathMetrics toolbox for Matlab, as outlined by Noto et al. (2018). This toolbox allowed us to identify key points in the breathing cycle, specifically the inhalation and exhalation onsets and peaks. These points of interest were then used to create a continuous measure of breathing phase spanning the full breathing cycle (360°). This was achieved through linear interpolation: from 0° to 90° for the breathing samples between inhalation onset and inhalation peak, 90° to 180° from inhalation peak to exhalation onset, 180° to 270° from exhalation onset to exhalation peak and 270° to 360° from exhalation peak to the subsequent inhalation onset. For subsequent analyses, aimed at detecting variations in pupil size across the breathing cycle, we divided the breathing cycle into 18 equally spaced bins, each spanning 20°, allowing us to compare pupil size during specific parts of the breathing cycle across participants. Inhalations and exhalations lasting less than 500 ms or more than 6000 ms were considered invalid (on average, 7.15% of breathing data were considered invalid in this manner). Finally, the breathing data were down-sampled to 60 Hz to match the sampling frequency of the pupil data for Experiments 1–4, and down-sampled to 100 Hz for Experiment 5.

## Data analysis

To investigate whether pupil size varied across the breathing cycle in a statistically significant manner, we calculated a two-way (breathing phase × breathing route) repeated measures ANOVA comparing the average normalized pupil size values for each breathing bin and each breathing route. Since we also hypothesized that pupil size might be larger during mouth-breathing than nose-breathing, and that this effect might only establish itself over time, we conducted an additional two-way (time × breathing route) repeated measures ANOVA for Experiments 1 and 2. For this, we divided the 5 min pupil size recordings into 18 evenly spaced time bins, and compared the average absolute pupil size for each time bin and each breathing route. Furthermore, for Experiment 4, we calculated a two-way (breathing phase × breathing condition) repeated measures ANOVA comparing the average normalized pupil size values for each breathing bin and each breathing condition. And, for Experiment 5, two one-way repeated measures ANOVAs were calculated to see whether breathing phase significantly affected pupil size, separately for participants with and without olfactory bulbs.

For each ANOVA we performed Mauchly's test to check whether the assumption of sphericity was violated for the main effects or their interaction. When the assumption of sphericity was violated, we applied a Greenhouse–Geisser correction. In the one case where sphericity could not be assessed due to a singular sum of squares and products matrix, we performed the non-parametric Friedman test instead.

To determine whether the mean normalized pupil size for each of the 18 breathing bins differed significantly from zero, we conducted a non-parametric permutation test. This approach, which deviated from our pre-registered permutation test, was adopted based on reviewer feedback to avoid parametric assumptions about the data distribution and to provide robust control of the family-wise error rate across multiple comparisons.

For each breathing bin, we computed the one-sample *t*-statistic as:

$$t = \frac{\bar{x}}{s/n}, \qquad (1)$$

where $\bar{x}$ is the mean normalized pupil size, $s$ is the standard deviation and $n$ is the number of observations in the breathing bin.

The null hypothesis assumed that the mean normalized pupil size for each breathing bin was zero. To generate the null distribution, we performed 10,000 permutations for each breathing bin. During each permutation, the signs of the observations within the breathing bin were randomly flipped, preserving the magnitude and variability of the original data while simulating a dataset consistent with the null hypothesis.

To account for multiple comparisons across the 18 breathing bins, we used a maximum-statistic approach. For each permutation, we computed the *t*-statistics for all breathing bins and recorded the maximum absolute *t*-statistic across breathing bins. This procedure generated a null distribution of maximum test statistics, which was used to control the family-wise error rate.

For each breathing bin, the observed *t*-statistic was compared with the null distribution of maximum *t*-statistics. The family-wise error-corrected *P*-value for each breathing bin was calculated as the proportion of permuted maximum *t*-statistics greater than or equal to the observed absolute *t*-statistic.

For the nose- and mouth-breathing conditions, we had an average of 35,879 and 34,381 observations, respectively, within the 20° breathing bins when combining the data of all participants in Experiment 1, an average of 44,027 and 41,440 observations per breathing bin in Experiment 2, and an average of 884,721 and 847,209 observations per breathing bin in Experiment 3. For Experiment 4, we had an average of 53,836, 53,772 and 55,801 observations per breathing bin for normal breathing, and controlled slow and fast breathing, respectively. And for Experiment 5, we had an average of 85,948 and 236,943 observations per

breathing bin for participants with and without olfactory bulbs, respectively.

To examine how pupil size changes throughout the breathing cycle, we calculated the pupil size derivative for each phase bin. First, we determined the change in pupil size between consecutive observations across the entire recording. Next, for each breathing cycle, we summed these changes within each phase bin, resulting in a value representing pupil size change for each bin in that cycle. Finally, we averaged these values across all breathing cycles to obtain the pupil size derivative for each phase bin.

The ANOVAs were done in JASP (JASP Team, 2024) and Matlab. Figures were created in Matlab and aesthetically modified in Inkscape (Inkscape Project, 2018). To show how pupil size changes across the breathing cycle, we created polar plots, polar histograms and calculated the circular mean direction of individuals' pupil sizes across the breathing cycle using the native *polarhistogram.m* and *polarplot.m* native Matlab functions, as well as the *circ_mean.m* function of the Circular Statistics Toolbox for Matlab (Berens, 2009). The *circ_mean.m* function uses the following formula to calculate the mean direction for circular data:

$$direction = angle\left(sum\left(weights * (i * radians)^2\right)\right), \quad (2)$$

where the *weights* correspond to the average normalized pupil size for each breathing bin, and the *radians* correspond to the angle of each breathing bin in radians. The *angle* function returns the phase angle, in radians, of a complex number. In addition, we used a Rayleigh test to check whether the mean vector direction of the participants' pupil sizes was uniformly distributed over the breathing cycle, making use of the *circ_rtest.m* function of the Circular Statistics Toolbox for Matlab (Berens, 2009).

## Results

### General

The aggregated data from our five experiments (collapsed across all conditions), comprising 194 unique participants, showed a robust pattern: pupil size was smallest around the onset of inhalation and largest during exhalation (see Fig. 1*A* and *B*). Pupil dilatation occurred predominantly throughout inhalation and the early phase of exhalation, while constriction was primarily confined to the latter part of exhalation (Fig. 1*C*). This pattern was consistent across all five experiments. Below, we detail the findings from each experiment individually.

### Experiment 1 – pupil size at rest

In Experiment 1, we measured nasal and oral respiration and pupil size in participants under dim conditions at rest, with a nearby fixation point and upright head position. This allowed us to determine the potential relationship between respiratory cycles and pupillary dynamics under simple and controlled conditions.

The average breathing frequency during Experiment 1 was 13.58 (SD = 3.58) and 14.27 (SD = 3.67) breaths/minute for nose- and mouth-breathing, respectively.

**Relative pupil size.** We conducted a two-way repeated measures ANOVA with the 38 complete recordings of Experiment 1, to investigate the effects of breathing phase and breathing route on pupil size. We found a significant effect of breathing phase ($F_{(3.77)} = 5.19$, $P < 0.001$, $\eta^2 = 0.079$, Greenhouse–Geisser corrected), but no significant effect of breathing route ($F_{(1)} = 0.45$, $P = 0.506$, $\eta^2 < 0.001$), nor a significant interaction effect ($F_{(4.59)} = 0.63$, $P = 0.664$, $\eta^2 = 0.006$, Greenhouse–Geisser corrected). To make use of all the recordings and not just those of the participants that had complete recordings for both nose- and mouth-breathing, we also conducted separate one-way ANOVAs for nose-breathing ($F_{(3.31)} = 3.82$, $P = 0.00900$, $\eta^2 = 0.087$, n = 41, Greenhouse–Geisser corrected), and for mouth-breathing ($F_{(4.63)} = 3.10$, $P = 0.0120$, $\eta^2 = 0.067$, $n = 44$, Greenhouse–Geisser corrected), both of which confirmed a significant effect of breathing phase. The Rayleigh test showed that the mean vector directions were not uniformly distributed for either nose- ($P = 0.00250$) or mouth-breathing ($P = .0133$). For a majority of participants, the mean vector direction of pupil size over the breathing cycle pointed towards exhalation for both nose- (proportion = 0.73) and mouth-breathing (proportion = 0.66; see Fig. 2*A*). In line with this, average pupil size was largest around mid-exhalation and smallest around inhalation onset (see Fig. 2*B*). The averaged range of pupil size change during the pupil response was 0.13 *Z*-scores and 0.12 *Z*-scores during nose- and mouth-breathing, respectively (see Fig. 2*B*).

The permutation testing revealed that 17/18 phase bins were significantly different from zero for nose- and mouth-breathing, respectively, with 300/320–160° showing significantly smaller pupils and 160/180° – 300/320° showing significantly larger pupils. For the nose breathing condition, the bin spanning 160–180° was not significant with a *P*-value of 0.0628. All other bins were significant with *P*-values <0.001. For the mouth breathing condition, all bins were significant with *P*-values <0.001.

To highlight the difference between pupil size and changes in pupil size, we also plotted the change in pupil

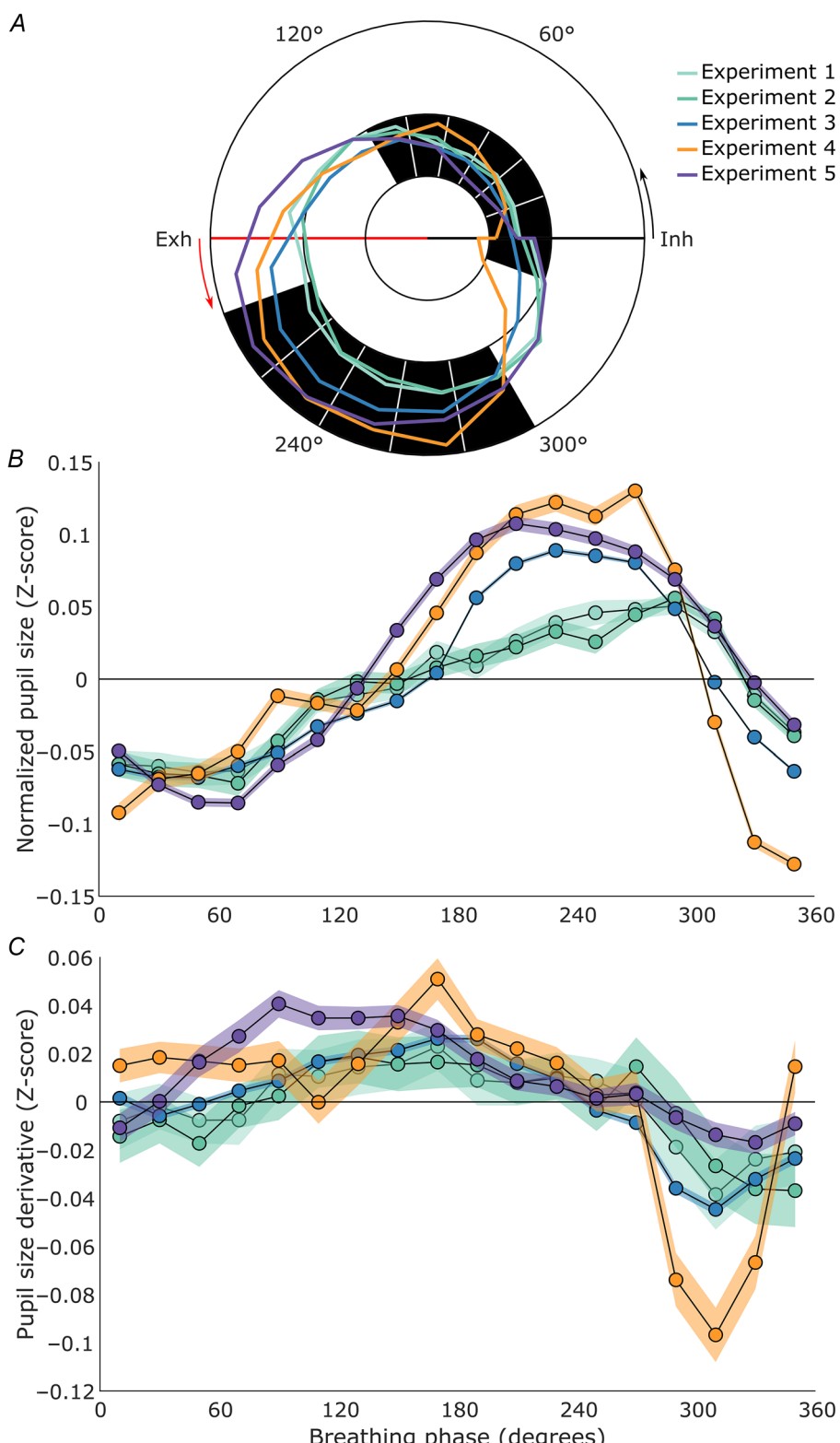

**Figure 1. Pupil response during the breathing cycle across the five experiments**
*A*, polar plot of averaged normalized pupil size over the course of the breathing cycle for each experiment. 0° (Inh) marks the onset of inhalation, 90° marks peak inhalation, 180° (Exh) marks the onset of exhalation and 270° marks peak exhalation. The inner black circle indicates a pupil size *Z*-score of −0.1, the middle black circle indicates a pupil size *Z*-score of 0, and the outer black circle indicates a pupil size *Z*-score of 0.15. The black shading indicates a phase

bin where the pupil size is significantly larger (outer ring) or smaller (inner ring) for all experiments. *B*, the line plot depicts the average normalized pupil size for each phase bin. Zero degrees on the *x*-axis corresponds to inhalation onset, 90° marks peak inhalation, 180° corresponds to exhalation onset, and 270° marks peak exhalation. The *y*-axis shows pupil size in *Z*-scores. The horizontal zero line represents the average normalized pupil size for each experiment. The shaded areas in all line plots represent the 95% confidence intervals. *C*, the line plot depicts the average change in normalized pupil size for each phase bin. The *x*-axis shows the breathing phase, and the *y*-axis shows the change in normalized pupil size (derivative) in *Z*-scores. All values above the zero line indicate dilating pupils, and all values below the zero line indicate constricting pupils. Data for Experiments 1–5 consist of $n = 47$, 51, 107, 53 and 34, respectively. [Colour figure can be viewed at wileyonlinelibrary.com]

size over the course of the breathing cycle (see Fig. 2*C*). This plot revealed that while the majority of pupil size dilatation occurred during inhalation and the majority of pupil constriction occurred during exhalation, the two processes were not strictly isolated to inhalation and exhalation, respectively.

**Absolute pupil size.** When looking at absolute pupil size changes over time ($n = 30$), we found a significant effect of time ($F_{(4.36)} = 3.97$, $P = 0.00400$, $\eta^2 = 0.048$, Greenhouse–Geisser corrected), but no significant effect of breathing route ($F_{(1)} = 0.01$, $P = 0.939$, $\eta^2 < 0.001$), nor a significant interaction effect ($F_{(4.69)} = 0.44$, $P = 0.810$, $\eta^2 = 0.003$, Greenhouse–Geisser corrected, see Fig. 2*D*). The decrease in pupil size over time might indicate a decrease in arousal as participants became more relaxed over the course of the recording.

### Experiment 2 – replication of pupil size at rest

To ensure the robustness of the findings from Experiment 1, which was exploratory, we performed a direct replication in Experiment 2 in an independent group of participants, but while keeping the experimental conditions the same.

During Experiment 2 the average breathing frequency was 12.26 (SD = 3.91) breaths/minute and 11.06 (SD = 4.35) breaths/minute for nose- and mouth-breathing, respectively.

**Relative pupil size.** Just as for Experiment 1, we found a significant effect of breathing phase ($F_{(5.04)} = 5.18$, $P < 0.001$, $\eta^2 = 0.064$, Greenhouse–Geisser corrected), but no significant effect of breathing route ($F_{(1)} = 3.30$, $P = 0.076$, $\eta^2 = 0.001$), nor a significant inter-action effect ($F_{(5.32)} = 0.87$, $P = 0.508$, $\eta^2 = 0.008$, Greenhouse–Geisser corrected), based on 44 complete recordings. The results from the one-way repeated measures ANOVAs showed a significant effect of breathing phase for nose-breathing ($F_{(5.79)} = 5.33$, $P < 0.001$, $\eta^2 = 0.102$, $n = 48$, Greenhouse–Geisser corrected), and for mouth-breathing ($F_{(4.60)} = 3.18$, $P = 0.0110$, $\eta^2 = 0.065$, $n = 47$, Greenhouse–Geisser corrected). The Rayleigh test showed that the mean vector directions were not uniformly distributed

for both nose- ($P = 0.00730$) and mouth-breathing ($P = 0.00300$). Again, we found that for the majority of the participants, the mean vector direction of pupil size over the breathing cycle pointed towards exhalation for both nose- (proportion = 0.71) and mouth-breathing (proportion = 0.72; see Fig. 2*A*). In line with this, on average, the pupil size was largest around mid-exhalation and smallest around inhalation onset (see Fig. 2*B*). The averaged range of pupil size change during the pupil response was 0.14 *Z*-scores and 0.13 *Z*-scores during nose- and mouth-breathing, respectively (see Fig. 2*B*).

The permutation testing revealed that 11 and 14 phase bins were significantly different from zero for nose- and mouth-breathing, respectively, with 320/340–100/120° showing significantly smaller pupils, and 180/260–320° showing significantly larger pupils. Specifically, for the nose-breathing condition, the bins spanning 120–220°, 240–260° and 320–340° were not significant, with *P*-values of 0.241, 0.999, 0.999, 1.000, 1.000, 0.956 and 0.289, respectively. Furthermore, the bin spanning 120-140° was significant at a *P*-value of 0.0110. All other bins were significant with *P*-values <0.001. For the mouth-breathing condition, the bins spanning 100–180° were not significant, with *P*-values of 1.000, 0.782, 1.000 and 0.496, respectively. All other bins were significant with *P*-values <0.001.

Similar to Experiment 1, during Experiment 2 the majority of pupil size dilatation occurred during inhalation and the majority of pupil constriction occurred during exhalation (see Fig. 2*C*).

**Absolute pupil size.** Assessing absolute pupil size changes over time ($n = 41$) again showed a significant effect of time ($F_{(2.95)} = 4.64$, $P = 0.00400$, $\eta^2 = 0.038$, Greenhouse–Geisser corrected), with pupil size decreasing over the course of the recording, but no significant effect of breathing route ($F_{(1)} = 1.18$, $P = 0.284$, $\eta^2 = 0.012$), nor a significant inter-action effect ($F_{(3.98)} = 0.76$, $P = 0.552$, $\eta^2 = 0.004$, Greenhouse–Geisser corrected, see Fig. 2*D*).

### Experiment 3 – pupil size during visual task

In Experiment 3, we determined whether results from Experiments 1 and 2, in which participants did not

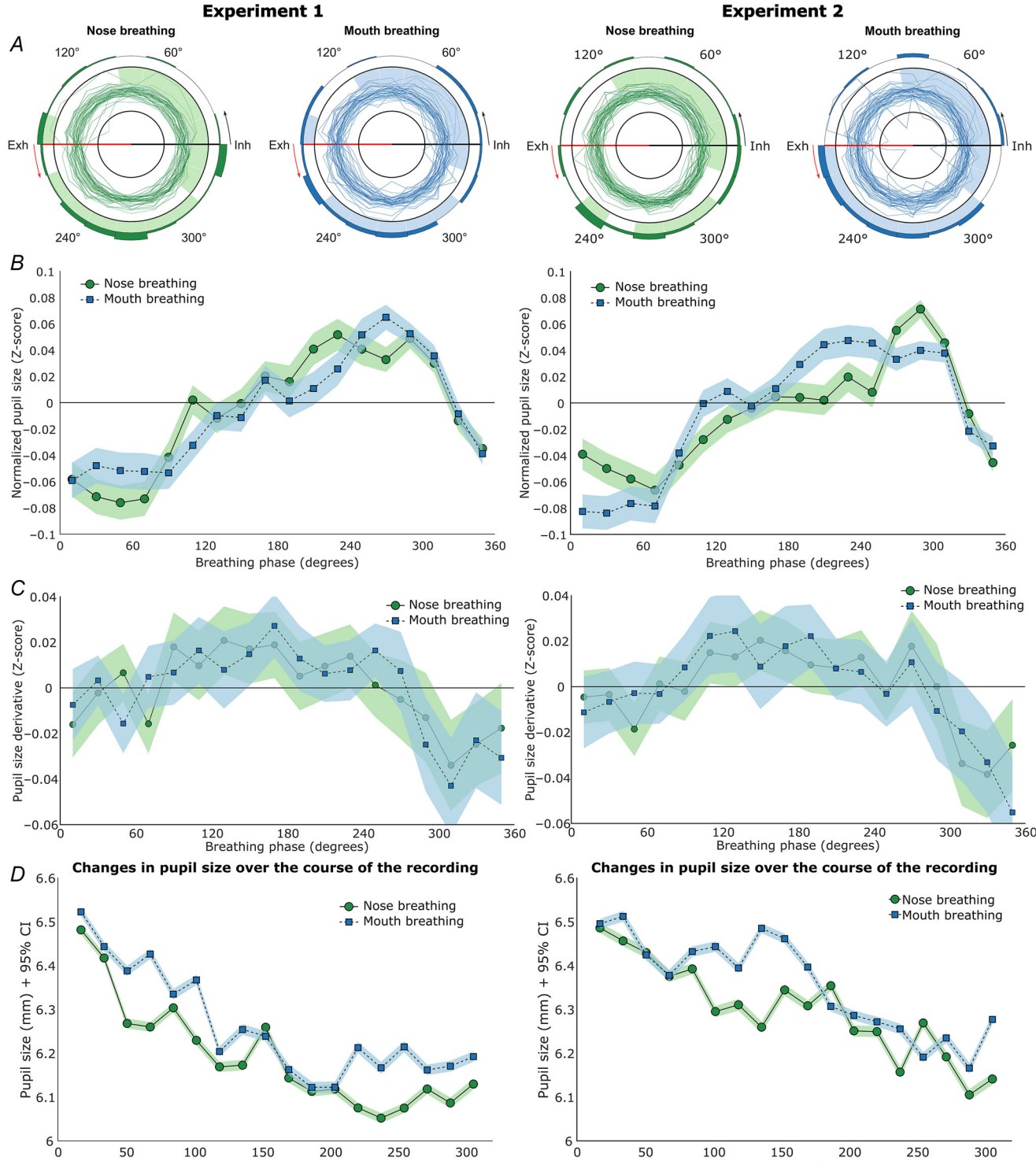

**Figure 2. Pupil size at rest**

*A*, polar plots of participants' pupil sizes over the course of the breathing cycle. 0° (Inh) marks the onset of inhalation, 90° marks peak inhalation, 180° (Exh) marks the onset of exhalation, and 270° marks peak exhalation. The individual dark green/dark blue lines show the normalized pupil size values of individual participants during nose- and mouth-breathing, respectively. The inner black circle indicates a pupil size *Z*-score of −0.5, and the outer black circle indicates a pupil size *Z*-score of 0.5. The light green/light blue shading indicates a phase bin that is significantly different from zero. A shaded bin outside the outer black circle indicates larger pupils, and a

shaded bin on the inside of the outer black circle indicates smaller pupils. The polar histogram (dark green/dark blue bins) on the outside of the polar plot shows the proportion of circular mean directions pointing towards a certain breathing phase bin (the histogram bins add up to a total of 1). The left side is based on the normalized pupil size data from Experiment 1 (n = 41 and 44 for nose- and mouth-breathing, respectively), and the right side is based on the normalized pupil size data from Experiment 2 (n = 48 and 47 for nose- and mouth-breathing, respectively). *B*, line plots depicting the average normalized pupil size for each phase bin. Zero degrees on the *x*-axis corresponds to inhalation onset, 90° marks peak inhalation, 180° corresponds to exhalation onset and 270° marks peak exhalation. The *y*-axis shows pupil size in *Z*-scores. The horizontal zero line represents the average pupil size for each individual. Pupil size during nose-breathing is depicted by the green circles connected by a continuous line, and pupil size during mouth-breathing is depicted by the blue squares connected by a dashed line. The shaded areas in all line plots represent the 95% confidence intervals. *C*, the line plot depicts the average change in normalized pupil size for each phase bin. The *x*-axis shows the breathing phase, and the *y*-axis shows the change in normalized pupil size (derivative) in *Z*-scores. All values above the zero line indicate dilating pupils, and all values below the zero line indicate constricting pupils. *D*, the line plots depict the average absolute pupil size for each time point. The *x*-axis shows the recording time in seconds. The *y*-axis shows pupil size in millimetres. The left plot is on the data from Experiment 1 (n = 35 and 39 for nose- and mouth-breathing, respectively), and the right plot is based on the data from Experiment 2 (n = 47 and 43 for nose- and mouth-breathing, respectively). [Colour figure can be viewed at wileyonlinelibrary.com]

perform any task, persisted during active visual perception under otherwise the same conditions. So, we measured breathing and pupil size while participants performed a three-alternative forced choice visual detection task.

For Experiment 3, the average breathing frequency was 13.94 (SD = 4.26) and 13.73 (SD = 4.78) breaths/minute for nose- and mouth-breathing, respectively.

We statistically confirmed that the effect of breathing phase held true during the visual task as well (F(3.03) = 80.44, $P < 0.001$, $\eta^2 = 0.365$, Greenhouse–Geisser corrected). Furthermore, there was again no significant effect of breathing route (F(1) = 3.43, $P = 0.067$, $\eta^2 < 0.001$), nor a significant interaction effect (F(4.01) = 1.00, $P = .410$, $\eta^2 = 0.002$, Greenhouse–Geisser corrected), as shown by a two-way ANOVA with 98 complete recordings.

Now that the recording duration was longer (~45 min rather than 5 min), an even more consistent pattern emerged (see Fig. 3A). This convergence became especially clear when assessing the average mean vector direction for all participants, which for both nose- (n = 103) and mouth-breathing (n = 102), pointed towards early exhalation (see Fig. 3A). The proportion of participants that had their mean vector direction of pupil size point towards exhalation increased to 0.87 and 0.84 for nose- and mouth-breathing, respectively, and the Rayleigh test confirmed that the mean vector directions were not uniformly distributed for both nose- ($P < 0.001$) and mouth-breathing ($P < 0.001$).

The combined average of all participants during the visual task again showed that pupil size tended to peak during exhalation and had a minimum around inhalation onset for both nose- and mouth-breathing (see Fig. 3B). The averaged range of pupil size change during the pupil response was 0.16 *Z*-scores and 0.15 *Z*-scores during nose- and mouth-breathing, respectively (see Fig. 3B).

The permutation testing revealed that 17/18 phase bins were significantly different from zero for nose- and mouth-breathing, respectively, with 300/320–160° showing significantly smaller pupils and 160/200–300/320° showing significantly larger pupils. For the nose-breathing condition, the bins spanning 140–160° and 180–200° were not significant, with *P*-values of 1.000 and 0.323, respectively. Furthermore, the bins spanning 100–140° and 160–180° were significant, with *P*-values of 0.00210, 0.0199 and 0.0253, respectively. All other bins were significant, with *P*-values <0.001. For the mouth-breathing condition, the bins spanning 120–160° were not significant, with *P*-values of 1.000 and 0.791, respectively. Furthermore, the bins spanning 160–200° were significant, with *P*-values of 0.00620 and 0.00610, respectively. All other bins were significant, with *P*-values <0.001.

Furthermore, during the visual task, the majority of inhalation consisted of pupil size dilatation, and pupil constriction predominantly occurred during exhalation (see Fig. 3C).

### Experiment 4 – pupil size during controlled slow and fast breathing

In Experiment 4, we tested if the results from the previous experiments also persisted during controlled breathing (fast and slow) under more brightly lit conditions and with a distant point of focus. Because controlled breathing is known to activate different brain regions from passive breathing, and light intensity and fixation distance are known to influence pupil size, these factors have the potential to influence the effect that breathing phase has on pupil size (for review see Mathôt, 2018; Trevizan-Baú et al., 2024).

For Experiment 4, average breathing frequency was 13.43 (SD = 4.46), 7.96 (SD = 1.31) and 15.33 (SD = 0.51) breaths/minute for normal, slow and fast breathing, respectively.

Just as for Experiments 1–3, the effect of breathing phase was significant ($F_{(4.28)} = 32.27$, $P < 0.001$, $\eta^2 = 0.187$, Greenhouse–Geisser corrected). However, there was also a significant effect of breathing condition ($F_{(2)} = 10.31$, $P = 0.001$, $\eta^2 < 0.005$), and a significant interaction effect ($F_{(9.08)} = 5.78$, $P < 0.001$, $\eta^2 = 0.051$, Greenhouse–Geisser corrected), based on 51 complete recordings. Again, we found that for a majority of participants, the mean vector direction of pupil size over the breathing cycle pointed towards exhalation for normal (proportion = 0.83), slow (proportion = 0.61)

and fast breathing (proportion = 0.75; see Fig. 4*A*), and the Rayleigh test confirmed that the mean vector directions were not uniformly distributed for normal breathing ($P < 0.001$), slow-paced breathing ($P < 0.001$), or fast-paced breathing ($P < 0.001$). Similarly, on average, pupil size was again largest around mid-exhalation and smallest around inhalation onset for all conditions (see Fig. 4*B*).

The permutation testing revealed that 17/13/17 phase bins were significantly different from zero for normal, slow and fast breathing, respectively, with

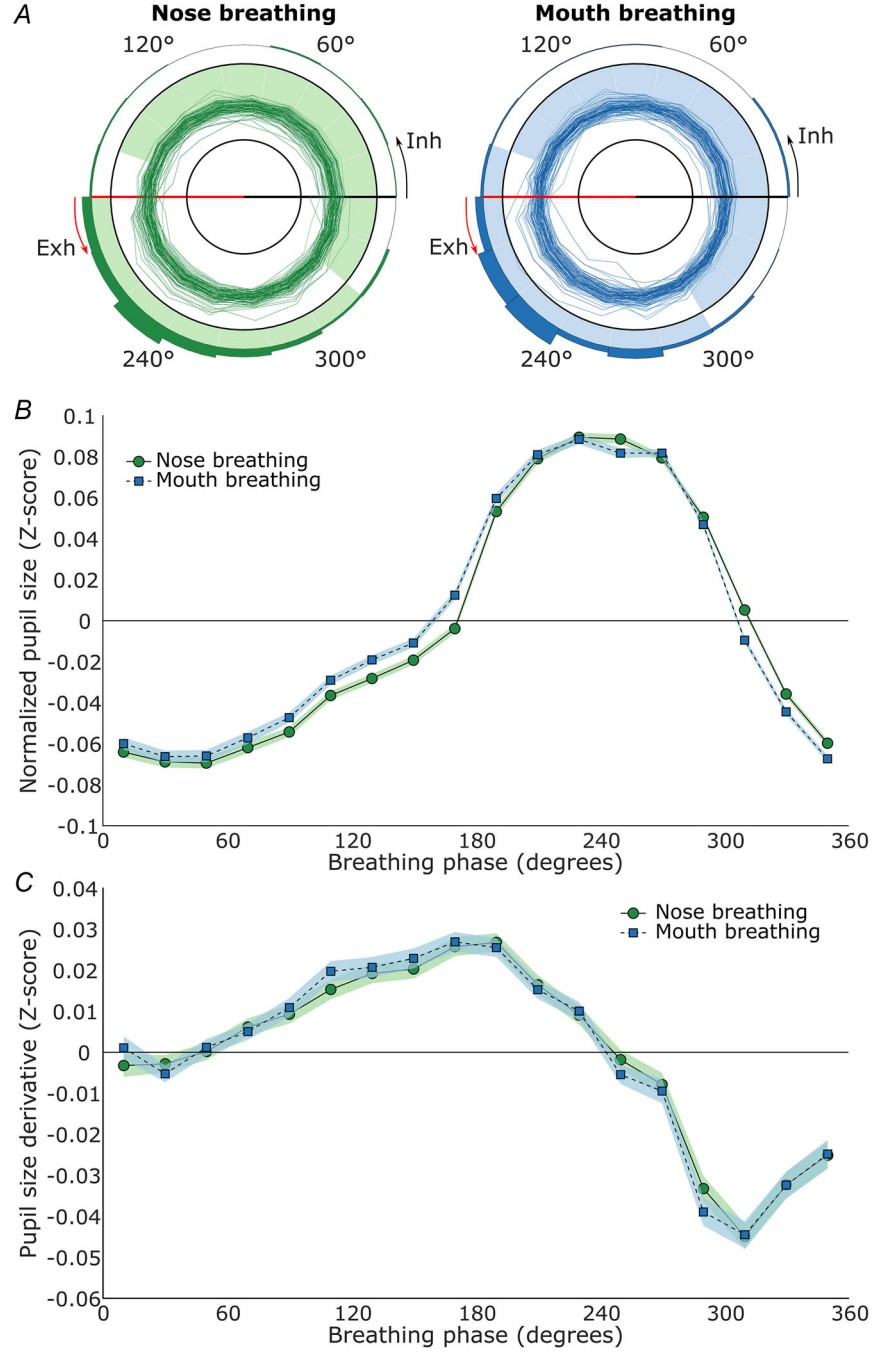

**Figure 3. Pupil size during visual task**
Results from Experiment 3 (*n* = 103 and 102 for nose- and mouth-breathing, respectively). *A*, polar plots of participants' pupil sizes over the course of the breathing cycle (see Fig. 2*A* legend for a more detailed description). *B*, the line plots depict the average normalized pupil size for each phase bin. Zero degrees on the *x*-axis corresponds to inhalation onset, 90° marks peak inhalation, 180° corresponds to exhalation onset and 270° marks peak exhalation (see Fig. 2*B* legend for a more detailed description). The shaded areas represent the 95% confidence intervals. *C*, the line plots depict the average change in normalized pupil size for each phase bin. The *x*-axis shows the breathing phase and the *y*-axis shows the change in normalized pupil size (derivative) in *Z*-scores. All values above the zero line indicate dilating pupils, and all values below the zero line indicate constricting pupils. [Colour figure can be viewed at wileyonlinelibrary.com]

320–20° showing significantly smaller pupils, and 160–280° showing significantly larger pupils for all three breathing conditions (see Fig. 4*A*). Specifically, for the normal breathing condition, all breathing bins were significant, with *P*-values <0.001 except for the bin spanning 140–160° which had a *P*-value of 1.000. For the slow-breathing condition, the bins spanning 20–80°, 100–120° and 280–300° were not significant, with *P*-values of 0.912, 1.000, 0.476, 0.511 and 0.583, respectively. Furthermore, the bin spanning 120–140° was significant, with a *P*-value of 0.007. All other bins were significant,

with *P*-values <0.001. For the fast-breathing condition, 17 breathing bins were significant, with *P*-values <0.001, except for the bins spanning 100–120°, and 140–180°, which were significant, with *P*-values of 0.0211 and 0.0325, respectively. Furthermore, the breathing bin spanning 80–100° was not significant, with a *P*-value of 0.458.

Furthermore, we again found that the majority of pupil size dilatation occurred during inhalation and the majority of pupil constriction occurred during exhalation (see Fig. 4*C*). However, the pattern of pupil dilatation

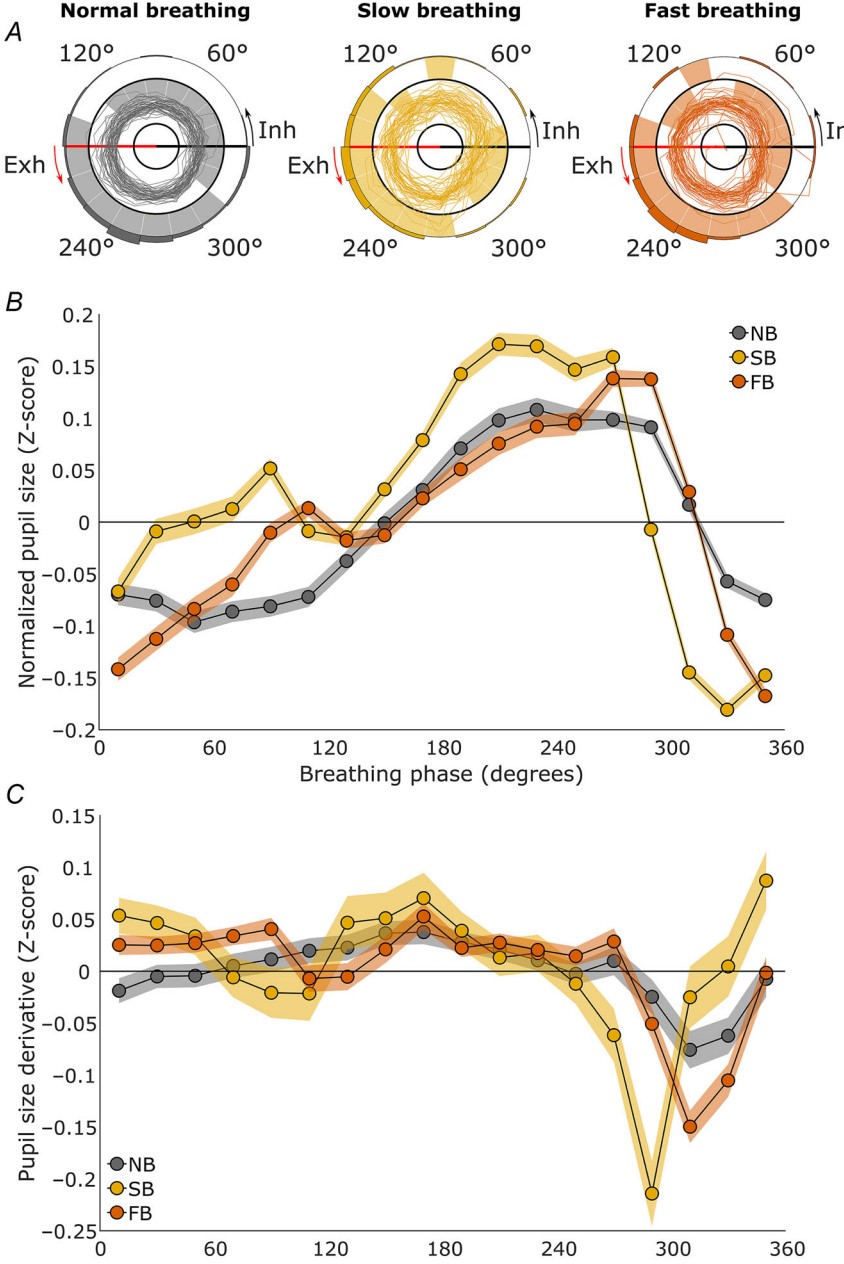

**Figure 4. Pupil size during paced breathing**
Results from Experiment 4 (*n* = 53, 53 and 55 for normal, slow, and fast breathing, respectively). *A*, polar plots of participants' pupil sizes over the course of the breathing cycle (see Fig. 2*A* legend for a more detailed description). *B*, the line plots depict the average normalized pupil size for each phase bin. Zero degrees on the *x*-axis corresponds to inhalation onset, 90° marks peak inhalation, 180° corresponds to exhalation onset and 270° marks peak exhalation (see Fig. 2*B* legend for a more detailed description). The shaded areas represent the 95% confidence intervals. *C*, the line plots depict the average change in normalized pupil size for each phase bin. The *x*-axis shows the breathing phase, and the *y*-axis shows the change in normalized pupil size (derivative) in *Z*-scores. All values above the zero line indicate dilating pupils, and all values below the zero line indicate constricting pupils, the shaded areas represent 95% confidence intervals. NB, normal breathing; FB, fast paced breathing; SB, slow paced breathing. [Colour figure can be viewed at wileyonlinelibrary.com]

and constriction differed slightly for the paced breathing condition as compared with the normal breathing.

In contrast to Experiments 1–3, we also had access to accurate absolute pupil size measurements for Experiment 4. This allowed a more concrete evaluation of the effect size of the pupil response. The average pupil size was 4.53 mm (SD = 0.89 mm), with an average range of 0.11 mm or 0.20 Z-scores over the breathing cycle during normal breathing, 4.67 mm (SD = 1.00 mm) with an average range of 0.23 mm or 0.35 Z-scores during controlled slow breathing, and 4.61 mm (SD = 0.96 mm) with an average range of 0.18 mm or 0.31 Z-scores during controlled fast breathing (see Fig. 4B).

### Experiment 5 –pupil size in participants with and without olfactory bulbs

Finally, in Experiment 5, we tested whether the effect of respiration phase on pupil size persisted in the absence of the olfactory bulb respiratory oscillator during nasal respiration. Since the olfactory bulb is known to generate important global respiratory-coupled oscillations, its absence might influence the effect of respiration phase on pupil size (Karalis & Sirota, 2022; Kay et al., 2009; Tort et al., 2018; Zelano et al., 2016). To investigate this, we used a lesion-type model involving participants born without olfactory bulbs, a rare disorder causing isolated congenital anosmia, but otherwise normal health, alongside control participants. Both groups were tested during visual and auditory tasks while lying down.

During Experiment 5, the average breathing frequency was 15.19 (SD = 3.54) and 14.87 (SD = 4.07) breaths/minute for participants with normosmia and isolated congenital anosmia, respectively.

For the participants with normosmia the assumption of sphericity could not be assessed because the sum of squares and products matrix was singular. Therefore, we performed a Friedman test instead. Just as for Experiments 1–4, breathing phase significantly affected pupil size for participants with olfactory bulbs ($\chi 2(17) = 126.73$, $P < 0.001$, Kendall's W = 0.573), as well as participants without olfactory bulbs ($F(3.07) = 22.367$, $P < 0.001$, $\eta^2 = 0.528$, Greenhouse–Geisser corrected). Again, we found that for a majority of participants, the mean vector direction of pupil size over the breathing cycle pointed towards exhalation for participants with olfactory bulbs (proportion = 0.92), and participants without olfactory bulbs (proportion = 0.95; see Fig. 5A), and the Rayleigh test confirmed that the mean vector directions were not uniformly distributed for either participants with olfactory bulbs ($P < 0.001$) or without olfactory bulbs ($P < 0.001$). Similarly, on average, pupil size was again largest around mid-exhalation and smallest around inhalation onset for both groups (see Fig. 5B).

The averaged range of pupil size change during the pupil response was 0.22 Z-scores and 0.20 Z-scores for participants with and without olfactory bulbs, respectively (see Fig. 5B).

The permutation testing revealed that 17 phase bins were significantly different from zero for participants with and without olfactory bulbs, with 320/340–120/140° showing significantly smaller pupils, and 140–320° showing significantly larger pupils. Specifically, all bins were significant, with P-values <0.001, except for the breathing bin of participants with olfactory bulbs spanning 120–140°, which had a P-value of 1.000, and for the breathing bin of participants without olfactory bulbs spanning 120–140°, which had a P-value of 0.0250 and the bin spanning 320–340°, which had a P-value of 0.0818.

Assessing the change in pupil size revealed that most of the inhalation consisted of pupil dilatation and most of the pupil constriction occurred during the second half of exhalation for both groups (see Fig. 5C).

## Discussion

Across five experiments, our findings consistently show that pupil size is smallest around inhalation onset and largest during exhalation. This pattern holds under a wide range of conditions: whether breathing through the nose or mouth, at free or controlled rates, with slow or fast breathing, during both rest and active tasks, under bright and dim lighting, while focusing at near and far distances, in vertical and horizontal head positions and in the presence or absence of the olfactory bulb. We term this effect the pupillary respiratory-phase response (PRP response). This a novel type of pupil response, alongside the previously established pupillary light response, pupillary near response and psychosensory pupil response (Mathôt, 2018).

The observed changes in pupil size for the PRP response had an effect size ranging from 0.11 to 0.23 mm change on average across the respiratory cycle depending on the breathing condition (see Experiment 4), or ranging from 0.12 to 0.35 Z-scores across experiments (see Experiments 1–5). Considering that the human pupil can vary by up to *ca* 6 mm from its smallest (∼2 mm) to largest (∼8 mm) size under normal circumstances (Mathôt, 2018), this observed effect represents about 1.8–3.8% of the total range. This effect is too small to be readily apparent in individual breathing cycles, but it emerges when averaging pupil size over several breathing cycles. One way to contextualize this effect size is to compare it with the three other known pupil responses (Mathôt, 2018). The pupillary light response, which constricts the pupil in response to light, and the pupillary near response, which constricts the pupil during near fixation, can both

change the pupil by up to several millimetres. In contrast, the pupillary psychosensory response, which dilates the pupil in response to cognitive activity (e.g. arousal and mental effort), results in much smaller changes, generally only a few tenths of a millimetre. The PRP response is comparable in effect size to the pupillary psychosensory response. However, the PRP response is unique among pupil responses due to its cyclic nature, which involves both dilatation and constriction, its constant presence, and

its exclusively internal origin, making it the only known pupil response with these characteristics.

Notably, while inhalation is predominantly marked by pupil size dilatation, and the majority of pupil constriction occurs during exhalation, the two processes do not seem to be strictly isolated to inhalation and exhalation, respectively. This adds some nuance to previous assumptions, primarily based on Borgdorff's (1975) landmark study on cats, where pupil dilatation was

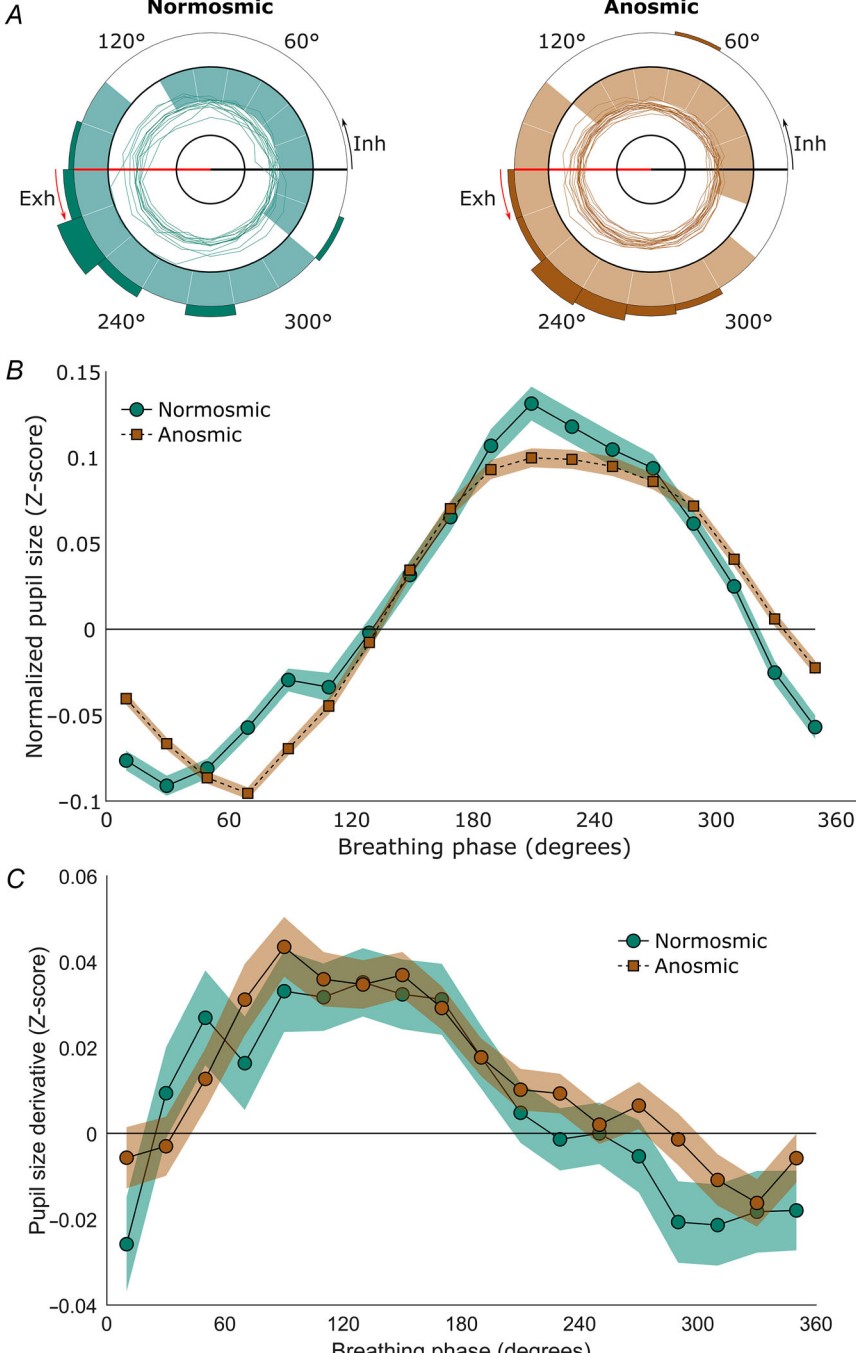

**Figure 5. Pupil size in participants with and without olfactory bulbs**
Results from Experiment 5 ($n = 13$ and 21 for participants with and without olfactory bulbs, respectively). *A*, polar plots of participants' pupil sizes over the course of the breathing cycle (see Fig. 2*A* legend for a more detailed description). *B*, the line plots depict the average normalized pupil size for each phase bin. Zero degrees on the *x*-axis corresponds to inhalation onset, 90° marks peak inhalation, 180° corresponds to exhalation onset and 270° marks peak exhalation (see Fig. 2*B* legend for a more detailed description). The shaded areas represent the 95% confidence intervals. *C*, the line plots depict the average change in normalized pupil size for each phase bin. The *x*-axis shows the breathing phase, and the *y*-axis shows the change in normalized pupil size (derivative) in *Z*-scores. All values above the zero line indicate dilating pupils, and all values below the zero line indicate constricting pupils. [Colour figure can be viewed at wileyonlinelibrary.com]

observed during inhalation and constriction during exhalation in lightly anaesthetized or tranquillized cats. However, this effect disappeared in fully awake or heavily anaesthetized cats. Despite Borgdorff (1975) acknowledging this limitation, his findings have been interpreted as evidence for a broader physiological mechanism. Additionally, although not explicitly stated, the figures in Borgdorff's (1975) paper depicted maximal dilatation during inhalation and minimal dilatation during exhalation, suggesting that pupils are largest during inhalation and smallest during exhalation. However, here we show conclusively that this is not the case and that in fact the effect is in the opposite direction.

While previous studies have suggested that respiration may influence various aspects of pupil function in humans, none have described an effect like the PRP response we observed. For example, breathing has been found to synchronize with pupil size changes during rest and tasks (Melnychuk et al., 2018, 2021). Also, it has been shown that pupil size fluctuated with the breathing cycle at rest, but this effect disappeared during a visual task (Nakamura et al., 2019). Additionally, breathing has been shown to influence the pupillary light reflex (Golenhofen & Petrányi, 1967; Kaulen et al., 1979) and the pupillary hippus (Kluger et al., 2024).

There are several reasons why the PRP response may have gone unnoticed until now. One possibility is that no previous studies have directly investigated it. Even if they had, the small sample sizes commonly used in earlier research (mean of 13 participants; Schaefer et al., 2022), the absence of pre-registered replication protocols and the use of short recording sessions would have made it difficult to detect. A critical factor in our ability to identify this effect was dividing the breathing cycle into 18 fine-grained bins, allowing for more precise measurement of pupil size changes throughout the respiratory cycle. This approach stands in contrast to prior studies, which predominantly relied on a binary inhalation *versus* exhalation division of the respiratory phase (Schaefer et al., 2022).

## Mechanisms underlying the PRP response

While our experiments do not allow for a direct inference about the neural mechanism controlling the PRP response, they offer indirect insights. This is because we manipulated critical factors known to influence the neural mechanisms underlying both breathing and pupil function. First, we can conclude that the mechanisms underlying the PRP response are distinct from the other three known pupil responses. The PRP response was present while keeping lighting conditions constant, at various levels of ambient luminance, and at different fixation points suggesting that its origin is distinct from the pupillary light response, and the pupillary near response, though it may interact with them. It is also distinct from the psychosensory pupil response as we observed the same PRP pattern at rest and during tasks that are mentally taxing. Therefore, we believe the PRP response is not caused by cognitive or arousal states, although it may interact with them.

**Insights from active *vs*. passive breathing.** During autonomous, passive breathing, the rhythm and motor pattern of respiration is generated by the respiratory centres in the pons and the medulla guided by the information relayed from chemo- and mechanosensors. In contrast, during controlled, volitional breathing, respiration is modulated by the cortex (for review see Trevizan-Baú et al., 2024). Additionally, during active expiration, expiratory pump muscles – particularly the internal intercostal and abdominal muscles – are engaged, which does not occur during passive expiration (De Troyer et al., 2005; Mortola, 2013; Welch et al., 2019). These changes represent potential neural mechanisms by which the observed pupil response could have been modified. However, as demonstrated in Experiment 4, neither active control over inhalation and exhalation, nor the pace of breathing diminish the PRP response, although these factors might influence its magnitude or modify its shape (see Experiments 1–4). Autonomous respiration involves active inspiration, commonly followed by post-inspiration and passive expiration (del Negro et al., 2018; Krohn et al., 2023). In this context, the most prevalent manner in which respiration can affect pupil size is through excitatory inputs from the preBötzC to the locus coeruleus, which are phase-locked to inspiration, leading to a respiration-regulated release of noradrenaline by the locus coeruleus (del Negro et al., 2018; Yackle et al., 2017). The locus coeruleus projects to the intermedio-lateral column, which in turn projects to the superior cervical ganglion which innervates the iris dilator muscle (Mathôt, 2018; Szabadi, 2018). This pathway may directly drive PRP responses. Interestingly, the dilatation could not have resulted from inhibition of the Edinger–Westphal nucleus' parasympathetic constriction pathway, as this mechanism is primarily associated with dilatation due to arousal and mental effort (Mathôt, 2018; Szabadi, 2018), and the PRP response remained unchanged during task performance.

**Insights from breathing route and from the absence of olfactory bulbs.** Past studies have shown that nasal and oral breathing differ at the behavioural and brain level primarily because nasal breathing engages additional neural circuits, particularly those related to the olfactory system (Arshamian et al., 2018; Fontanini & Bower, 2006; Heck et al., 2017; Karalis & Sirota, 2022; Kay

et al., 2009; Kluger et al., 2021; Kocsis et al., 2018; del Negro et al., 2018; Perl et al., 2019; Schaefer et al., 2024; Tort et al., 2018; Zelano et al., 2016). Specifically, nasal respiration engages the olfactory bulb, which generates global respiratory-coupled oscillations (Karalis & Sirota, 2022; Kay et al., 2009; Tort et al., 2018; Zelano et al., 2016). This means that there might be plausible differences between nasal and oral breathing on the PRP response. However, as we measured both nasal and oral breathing (Experiments 1–3) – to our knowledge, the first time this has been done in relation to pupil function – we could show that the response is independent of the breathing route. Also, in absolute pupil size measures, we showed that pupil size is not significantly affected by breathing route over time (Experiments 1 and 2). Finally, in Experiment 5, we demonstrated that 'removing' the olfactory bulbs during nasal respiration did not alter the results. This provides clear evidence that the PRP response is independent of the olfactory bulb, the only known region outside the brainstem that generates respiratory-coupled oscillations.

Collectively, the broad range of experimental conditions and manipulations strongly supports the conclusion that the PRP response is governed by early brainstem circuits. These circuits are consistent with those involved in autonomous respiration: preBötzC → locus coeruleus → noradrenaline release in response to inspiration → spinal intermediolateral column → superior cervical ganglion → iris dilator muscle. Even though we believe the preBötzC to be the driver of this effect, maximum pupil size (∼mid-exhalation) does not occur simultaneously with maximum preBötzC activity (∼inhalation onset). This indicates that it takes some time from the firing of the preBötzC to the maximal pupil dilatation by the constriction of the iris dilator muscle. This is unsurprising, but a more precise understanding of the temporal dynamics of the PRP response would be of interest for future studies.

In conclusion, we have demonstrated that pupil size is smallest around the onset of inhalation and largest during exhalation, with pupil dilatation occurring through most of inhalation and the early phase of exhalation, and pupil constriction occurring primarily during the latter part of exhalation. This response is robust across a wide range of conditions. Future behavioural experiments should investigate how the PRP response impacts visual perception. Previous studies suggest that smaller pupils improve visual acuity aiding in visual discrimination tasks, while larger pupils enhance the detection of faint stimuli (Mathôt & Ivanov, 2019). Our findings hint at the possibility that visual perception itself might cycle between optimizing for discrimination during inhalation and detection during exhalation within a single breath.

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

## Additional information

### Data availability statement

Pre-processed data and code are available here: https://osf.io/nk7zh/?view_only=cc85090d703e4c98be421e2bcfe31b6b. Raw data will be shared upon reasonable request.

### Competing interests

None declared.

### Author contributions

Conception and design of the work: A.A., M.S. Acquisition, analysis or interpretation of data for the work: A.A., J.N.L., M.L., M.S., S.M. Drafting the work or revising it critically for important intellectual content: A.A., J.N.L., M.L., M.S., S.M. All authors have read and approved the final version of this manuscript and agree to be accountable for all aspects of the work in ensuring that questions related to the accuracy or integrity of any part of the work are appropriately investigated and resolved. All persons designated as authors qualify for authorship, and all those who qualify for authorship are listed.

### Funding

This research was funded by the Swedish Research Council (VR 2018–01603) and (VR 2021–01311) awarded to A. Arshamian. In addition, M. Lundqvist was funded by ERC starting grant 949131.

### Acknowledgements

We would like to thank Sylvia Edwards, Erik Gustavsson, Emil Lejonklou and Reza Ebrahimian Baboukani for their help with the data collection.

### Keywords

breathing, human behaviour, human physiology, pupil constriction, pupil dilatation, pupil size, respiration, visual perception

## Supporting information

Additional supporting information can be found online in the Supporting Information section at the end of the HTML view of the article. Supporting information files available:

**Peer Review History**

