## [Peer Review History · The Journal of Physiology]

The Pupillary Respiratory-Phase Response: Pupil size is smallest around inhalation onset and largest during exhalation

Martin Schaefer, Sebastiaan Mathot, Mikael Lundqvist, Johan N Lundstrom, and Artin Arshamian

DOI: 10.1113/JP287205

Corresponding author(s): Martin Schaefer (martin.schaefer@ki.se)

The following individual(s) involved in review of this submission have agreed to reveal their identity: Sean Williams (Referee #3)

Review Timeline:

Submission Date:	02-Jul-2024
Editorial Decision:	05-Aug-2024
Revision Received:	26-Nov-2024
Editorial Decision:	20-Dec-2024
Revision Received:	13-Jan-2025
Accepted:	31-Jan-2025

Senior Editor: Harold Schultz

Reviewing Editor: Daniel Zoccal

Transaction Report:

Dear Dr Schaefer,

Re: JP-RP-2024-287205 "The Respiratory-Pupillary Phase Effect: Pupil size is smallest around inhalation onset and largest during exhalation" by Martin Schaefer, Sebastiaan Mathot, Mikael Lundqvist, Johan N Lundstrom, and Artin Arshamian

Thank you for submitting your manuscript to The Journal of Physiology. It has been assessed by a Reviewing Editor and by 2 expert referee and we are pleased to tell you that it is potentially acceptable for publication following satisfactory major revision.

REVISION CHECKLIST:

We look forward to receiving your revised submission.

Yours sincerely,

Harold Schultz
Senior Editor
The Journal of Physiology

REQUIRED ITEMS

- Author photo and profile. First or joint first authors are asked to provide a short biography (no more than 100 words for one author or 150 words in total for joint first authors) and a portrait photograph. These should be uploaded and clearly labelled together in a Word document with the revised version of the manuscript. See Information for Authors for further details.
- Your manuscript must include a complete Additional Information section, including competing interests; funding; author contributions and acknowledgements.
- Please upload separate high-quality figure files via the submission form.
- Please ensure that the Article File you upload is a Word file.
- Papers must comply with the Statistics Policy: https://jp.msubmit.net/cgi-bin/main.plex?form_type=display_requirements#statistics.

In summary:

- If $n \leq 30$, all data points must be plotted in the figure in a way that reveals their range and distribution. A bar graph with data points overlaid, a box and whisker plot or a violin plot (preferably with data points included) are acceptable formats.
- If $n > 30$, then the entire raw dataset must be made available either as supporting information, or hosted on a not-for-profit repository, e.g. FigShare, with access details provided in the manuscript.
- 'n' clearly defined (e.g. x cells from y slices in z animals) in the Methods. Authors should be mindful of pseudoreplication.
- All relevant 'n' values must be clearly stated in the main text, figures and tables.
- The most appropriate summary statistic (e.g. mean or median and standard deviation) must be used. Standard Error of the Mean (SEM) alone is not permitted.

- Exact p values must be stated. Authors must not use 'greater than' or 'less than'. Exact p values must be stated to three significant figures even when 'no statistical significance' is claimed.

- Please include an Abstract Figure file, as well as the Figure Legend text within the main article file. The Abstract Figure is a piece of artwork designed to give readers an immediate understanding of the research and should summarise the main conclusions. If possible, the image should be easily 'readable' from left to right or top to bottom. It should show the physiological relevance of the manuscript so readers can assess the importance and content of its findings. Abstract Figures should not merely recapitulate other figures in the manuscript. Please try to keep the diagram as simple as possible and without superfluous information that may distract from the main conclusion(s). Abstract Figures must be provided by authors no later than the revised manuscript stage and should be uploaded as a separate file during online submission labelled as File Type 'Abstract Figure'. Please also ensure that you include the figure legend in the main article file. All Abstract Figures should be created using BioRender. Authors should use The Journal's premium BioRender account to export high-resolution images. Details on how to use and access the premium account are included as part of this email.

- Please include a full title page as part of your main article (Word) file, which should contain the following: title, authors, affiliations, corresponding author name and contact details, keywords, and running title.

Reviewing Editor's comments:

Two experts in the field assessed the study by Schaefer and colleagues, which examined the dynamic changes in pupil size associated with breathing in humans. Both reviewers acknowledged the study's relevance and noted that its quality was enhanced by the use of a larger sample size, more detailed analyses of the respiratory cycle, different conditions (resting and a visual task), and different breathing routes (oral and nasal). The reviewers also provided technical and scientific observations that require the authors' attention. These include the need for a clearer distinction between this and previous studies in the literature, the inclusion of supplementary information in the main text, clarifications about data analyses, and an examination of the influence of controlled breathing on the parameters analyzed. Additionally, I would like to address some confusion encountered in the manuscript. Figures 1 and 2 depict pupil size over a respiratory cycle at rest and during a visual task, indicating that pupil size is smallest during inspiration and largest during expiration. However, Figure 3, which shows the average change in pupil size during a respiratory cycle, appears to demonstrate a different dynamic, with pupil size increasing during inspiration and decreasing during expiration. This inconsistency, along with the authors' statement that pupil size "increases and decreases during both breathing phases" (page 18), creates uncertainty about the study's main findings. Furthermore, the current definition of respiratory phase onset and peak appears to be misleading. Using airflow signals would be more appropriate for identifying the phase onset and peak flow rather than assuming that any point within the initial 25% of each phase corresponds to the onset and the late 25% to the peak. The study should also present and compare participants' breathing parameters (frequency and volume) under resting conditions and during the visual task. This is particularly important given that respiration-pupil size entrainment seems to be strengthened during the visual task. Additionally, due to the large sample size, sex differences in pupil size dynamics might be explored. Please also include the number of participants in each protocol in the results section and figure captions and clarify why heart rate was monitored during the experiments. The abstract section can be improved by including more detailed information about experiments (humans, number of participants), methods, experimental protocols, and main conclusions of the study. Lastly, while the speculation about connections between the preBötC and LC is intriguing, the references cited do not provide experimental evidence supporting the existence of noradrenergic projections from the preBötC to the LC, nor the presence of noradrenergic interneurons in the LC receiving preBötC inputs. Clarification and resolution of these issues are essential to ensure the study's conclusions are accurately represented and understood.

Senior Editor:

Comments to ensure the paper complies with the Statistics Policy:

A post-hoc statistical analysis should be used to confirm differences in pupil size during the respiratory cycle.

Comments to the Authors:

Thank you for submission of your research article to the Journal of Physiology for consideration. The article has been reviewed by experts in the field and found to require a major revision to address all of the concerns raised before a decision can be made for acceptability for publication. Please address all comments from the external referees and reviewing editor as well as addressing the list of requirements or publication in the journal including the statistical requirements.

Please begin the Methods with the section Ethical Approval, rather than Participants. Begin with: Ethical approval was obtained from the national Swedish Ethical Review Authority (Dnr 2020-00972) and all participants signed informed consent prior to participation. The study conformed to the standards set by the Declaration of Helsinki, except for the lack of registration in a database before recruitment of the first participant (clause 35).

Please incorporate all of the supplementary material into the manuscript. The Journal does not accept supplementary methods, results, figures and tables. There is no restriction on page length for the manuscript.

Please revise the abstract to be more informative of design and outcomes.

A post-hoc statistical analysis should be used to confirm differences in pupil size during the respiratory cycle.

Referee #1:

Please see attached [JP-RP-2024-287205_Referee 1 Review Attachment 1.pdf].

Referee #2:

In the present study, the authors examined relative pupil size during respiration during both rest and while actively engaged in a visual attention experiment. They found that pupil size tended to increase as people inhaled peaked around maximum inhalation, then pupil size decreased with exhalation.

I have only a few minor questions I would like to see addressed.

1) I am curious about several findings mentioned. And while I do not believe additional data are necessary, I believe they would be informative. For example, the authors mention a possible difference between controlled and passive breathing. This seems like an easy follow-up experiment to perform. I could imagine a set of competing hypotheses, too. One hypothesis would be that self-regulated, controlled breathing would amplify the effect of respiration on pupil size because it would recruit attention, and perhaps consequently, the LC-NE system. Another hypothesis is that passive, uncontrolled breathing is regulated by the brainstem (which houses the LC), and therefore controlling one's breathing might wrest control over respiration away from the brainstem. But I don't know enough about the physiology of the connection between the brain and the muscles of the lungs to know if that's plausible. I would therefore be curious to see a comparison of controlled vs. passive breathing. Otherwise, is there extant data on controlled vs. passive breathing on pupil size in people?

2) Another thing that struck my interest was the potential for regulation of the LC-NE system by respiration. Many meditation techniques, some of which have been successfully employed as interventions to improve attention, use controlled and focused breathing. I wondered whether one potential mechanism is a momentary, or perhaps lasting, regulation of the LC-NE system, which might have downstream benefits on attention. That is, can breathing be a way to exert "control" over the LC-NE system? Is there any existing data that might speak to this?

Otherwise, I found this to be a very well conducted set of experiments and a well written manuscript. This was a topic about which I knew almost nothing and it seems important that the results, which were robust and clear, violate the traditional wisdom.

END OF COMMENTS

Respiratory modulation of pupil diameter has been known for decades. However, few direct studies on this phenomenon in humans are available, and published observations suffer from methodological limitations. Schaefer et al. aim to fill this gap in knowledge by directly studying how pupil diameter changes with breathing phase in humans at rest and during a visual task and when subjects are asked to breathe through their nose or mouth. The authors find that pupil size is maximal during peak exhalation and minimal during inhalation onset. Further, this pattern of respiratory modulation of pupil size, which they term the Respiratory-Pupillary Phase Effect (RPPE), does not differ whether subjects are breathing through their nose or mouth, or whether subjects are at rest or engaged in a visual perception task.

The manuscript presents straightforward studies that rigorously characterize the effects of breathing on pupil size in humans. The larger sample size, rigorous study design, and smaller phase bins are significant improvements over previous studies. The results are high quality and are an important contribution to our understanding of these basic physiological processes, particularly because the studies are conducted in humans. The lack of effect of breathing route and the effect of engagement in a task on RPPE are also novel contributions to the field. While the manuscript is generally clear and the conclusions are supported by the data, a number of issues should be addressed that would improve the manuscript.

1) The authors present a key, erroneous assumption about the prevailing view as being that pupil size is maximal during inhalation and minimal during exhalation. While I was unable to access the pre-Borgdorff 1975 references, I do not find this explicit claim in Borgdorff or the other more recent cited references. While one may draw this inference from the figures in Borgdorff, citing specific publications that make this claim would strengthen this argument. It appears that the prevailing view described in many of these studies is that pupil size changes with breathing phase and, perhaps more specifically, that pupil dilates during inhalation and constricts with exhalation, which is consistent with Borgdorff and the results of this study. In my view, the significance of this manuscript would not be significantly diminished if statements related to the minimum/maximum claim in the key points, abstract, introduction, and discussion were tempered or removed, especially if this claim is not explicitly made in published literature.

2) While combining data from Experiments 1 and 2 is justified, the statistical descriptions that are currently in Supplementary Materials for the individual datasets should be incorporated into the Results of the main text to support grouping the datasets. Supplementary Figures and Tables can remain in Supplementary Materials, but should be individually/specifically referenced in the text. There are also sentences related to one-way repeated measures ANOVAs in Supplementary Materials, where it is unclear whether these are for relative or absolute pupil size and what the rationale is for this statistical comparison. These should be more clearly described and moved to the main Results as well.

3) It would be helpful to have a description of how the circular mean direction of individual pupil size is calculated in the Methods. In the results, is the “average mean vector direction” the same as the “circular mean direction”? Is the magnitude of the vector ignored? Is the minimal pupil size 180 degrees from the angle of the circular mean direction?

4) In the Discussion, it is mentioned that the changes in pupil size were “highly significant yet relatively small.” Is this statement based on normalized z-score or absolute measurements of pupil size (in mm)? Some description of effects sizes and/or absolute measurements in Results might help justify this statement.

5) I was confused by the data related to the changes in absolute pupil size (Figures S4 and S7). The 2x2 repeated measures ANOVA describes time as a variable, and the figure shows 0 to 300 s on the x-axis, which would correspond to the duration of the recording. However, the figure legend mentions phase bins, which would not make sense for breathing on this time scale. Either the data show a steady decrease in absolute pupil size over the 5 minute duration of the task or a maximum pupil size at the onset of inhalation (if x-axis is phase), which is inverted from the normalized data. Please clarify whether the data is showing absolute pupil size over the duration of the task or over cycle phase and discuss the results in the Discussion.

6) A minority of subjects appeared to have a different phase relationship between breathing and pupil size, i.e., maximum pupil size during inhalation, that may have changed when engaged in a visual task. If possible, this source of intersubject variability and the effect of engagement in a visual task should be analyzed and discussed.

7) In the paragraphs discussing possible mechanisms underlying RPPE (paragraphs 4 and 7 in Discussion), the multisynaptic, neuromodulatory pathways may be sources of latency that introduce a phase delay or alter the phase relationship between neural activity in preBötzing Complex and pupil dilation. This possibility should be discussed.

8) “Negro et al., 2018” should be “del Negro et al., 2018”.

26 November, 2024

Harold Schultz

Senior Editor

The Journal of Physiology

=====

Dear Prof. Schultz,

Attached is our revised manuscript for consideration for publication in *The Journal of Physiology*, with a new title "The Pupillary Respiratory-Phase Response: Pupil size is smallest around inhalation onset and largest during exhalation" (Re: JP-RP-2024-287205). We appreciate the thoughtful review and feedback, which we believe has significantly improved the manuscript.

The current manuscript structure, from the introduction to the discussion, has undergone substantial changes, partially in response to the reviewers' suggestions, and partially because of a series of new findings. Specifically, we have included two additional experiments. These new experiments greatly extend our findings and demonstrate that the original effects observed in the first three experiments are robust across various breathing conditions, external conditions known to influence pupil size, and immune to potential olfactory bulb oscillations.

In Experiment 4, we tested whether the results generalized to controlled breathing (fast and slow) at rest in brighter conditions with a more distant focus point. In Experiment 5, we examined whether the effect persisted in the absence of the olfactory bulb respiratory oscillator, using a human lesion-type model with healthy individuals with isolated congenital anosmia who were born without olfactory bulbs, an exceedingly rare condition, and controls.

Across diverse conditions—including nasal and oral breathing, free breathing and controlled breathing at slow and fast paces, during rest and tasks, in different head positions, under bright and dark conditions, with near and distant focus points, and with or without the olfactory bulb—our findings consistently show that pupil size is smallest around inhalation onset and largest during exhalation. Importantly, contrary to previous beliefs, both dilation and constriction occur during inhalation and exhalation.

We believe that this effect which we term the Pupillary Respiratory-Phase (PRP) response represents an additional known mechanism that influences pupil size, alongside the previously identified mechanisms: the pupillary light response, the pupillary near response, and the psychosensory pupil response (Mathôt, 2018). The discovery of the PRP response significantly enhances our understanding of the mechanisms underlying pupil dynamics.

We argue that the robustness of this phenomenon across diverse breathing conditions, along with its independence from the olfactory bulb, more clearly suggests that it is controlled by brainstem circuits.

We are confident that this revised manuscript will engage a much broader audience.

The following is a point-by-point response to the reviewers' comments. Our responses are marked in blue, and the changes in the manuscript are highlighted in yellow. We hope this revised submission will meet the reviewers' expectations and be suitable for publication in *The Journal of Physiology*.

On behalf of all the authors,

Martin Schaefer

=====

Editors' comments

Reviewing Editor's comments:

Two experts in the field assessed the study by Schaefer and colleagues, which examined the dynamic changes in pupil size associated with breathing in humans. Both reviewers acknowledged the study's relevance and noted that its quality was enhanced by the use of a larger sample size, more detailed analyses of the respiratory cycle, different conditions (resting and a visual task), and different breathing routes (oral and nasal). The reviewers also provided technical and scientific observations that require the authors' attention.

These include the need for a clearer distinction between this and previous studies in the literature, the inclusion of supplementary information in the main text, clarifications about data analyses, and an examination of the influence of controlled breathing on the parameters analyzed.

Response: The manuscript has undergone substantial changes at all levels. We have included two additional experiments where we demonstrate that the "Pupillary Respiratory-Phase response" or PRP response is robust during controlled breathing at various speeds, under different lighting conditions, and when fixating on a more distant location, in different head positions, and even without olfactory bulbs. We believe that this greatly strengthens the evidence for the existence of the PRP response and demonstrates its presence under a wide variety of experimental conditions.

We have now clarified how our study distinguishes itself from previous studies in the discussion. Furthermore, we have included the supplementary information in the main text. This has led to the remaking of all the figures and has also changed the structure of the manuscript. We have added additional clarification to our methods section as well as to the figure legends to make our data analysis more understandable.

Additionally, I would like to address some confusion encountered in the manuscript. Figures 1 and 2 depict pupil size over a respiratory cycle at rest and during a visual task, indicating that pupil size is smallest during inspiration and largest during expiration. However, Figure 3, which shows the average change in pupil size during a respiratory cycle, appears to demonstrate a different dynamic, with pupil size increasing during inspiration and decreasing during expiration. This inconsistency, along with the authors' statement that pupil size "increases and decreases during both breathing phases" (page 18), creates uncertainty about the study's main findings.

Response: The dynamic appears different in Figure 3 than in Figures 1 and 2 because it depicts the *change* in pupil size rather than the *state* of pupil size. During most of inhalation, pupil size is indeed increasing, however, there are some points at inhalation onset during which the pupil size derivative is negative, indicating a decrease in pupil size. During early exhalation the pupil size continues to increase (which is also indicated in Figures 1 and 2 where pupil size reaches its peak during exhalation), and then

during mid to late exhalation pupil size decreases as is indicated by the negative pupil size derivative. Therefore, we made the statement that pupil size "increases and decreases during both breathing phases". In an attempt to clarify this, we now write in the results:

To highlight the difference between pupil size and changes in pupil size, we also plotted the change in pupil size over the course of the breathing cycle (see Figure 2C). This plot revealed that pupil size increased and decreased during both inhalation and exhalation, but the majority of pupil size constriction occurred during the end of exhalation.

Please note that we changed all of the figures and therefore also the figure numbers.

Furthermore, the current definition of respiratory phase onset and peak appears to be misleading. Using airflow signals would be more appropriate for identifying the phase onset and peak flow rather than assuming that any point within the initial 25% of each phase corresponds to the onset and the late 25% to the peak.

Response: We clarify that airflow signals were indeed used to identify phase onsets and peak flows. See the following part of the methods section:

“To analyze the respiratory patterns of each participant, we employed the BreathMetrics toolbox for Matlab, as outlined by Noto et al. (2018). This toolbox allowed us to identify key points in the breathing cycle, specifically the inhalation and exhalation onsets and peaks. These points of interest were then used to create a continuous measure of breathing phase spanning the full breathing cycle (360 degrees). This was achieved through linear interpolation: from 0° to 90° for the breathing samples between inhalation onset and inhalation peak, 90° to 180° from inhalation peak to exhalation onset, 180° to 270° from exhalation onset to exhalation peak, and 270° to 360° from exhalation peak to the subsequent inhalation onset.”

If the editor could specify which part of the manuscript led to the understanding that we assumed “that any point within the initial 25% of each phase corresponds to the onset and the late 25% to the peak.” we will gladly rewrite it to make it clearer. Furthermore, we have now added that

“90° marks peak inhalation” and “270° marks peak exhalation”

to the Figure legends, to describe the division of the breathing cycle in more detail.

The study should also present and compare participants' breathing parameters (frequency and volume) under resting conditions and during the visual task. This is particularly important given that respiration-pupil size entrainment seems to be strengthened during the visual task.

Response: Unfortunately, our breathing setup does not allow us to make any accurate statements regarding breathing volume. For that you would need a closed system spirometer/ pneumotachograph. However, we have now included descriptions of participants' average breathing frequency + SD for each experimental condition. Regarding the strengthening during the visual task, we believe that this is not due to the task per se, but to the substantial increase in recording time which is almost an order of magnitude larger. When analyzing only a 5-minute subset of the 45-minute visual task data (i.e., similar to the rest condition), there is also more variability in the results, similar to the data recorded at rest.

With more data, the variance in the pupil size dynamics is averaged out and the PRP response becomes clearer.

Additionally, due to the large sample size, sex differences in pupil size dynamics might be explored.

Response: Based on the editor's suggestion, we explored potential sex differences in the PRP response at rest. From a visual inspection, there do not appear to be any notable differences in how pupil size changes during the breathing cycle between male and female participants (see Figure 1 below). Given that we did not hypothesize and pre-registered any potential sex difference and that our sample does not have an equal distribution of male and female participants, we feel it would be premature to explore this aspect further at this time, and we prefer not to include it in the current manuscript.

Figure 1 – Sex differences in pupil size dynamics over the breathing cycle. The left plot depicts nose breathing, the right plot mouth breathing. The x-axis depicts the breathing phase in degrees, 0 marks the onset of inhalation, 90 marks peak inhalation, 180 marks the onset of exhalation, and 270 marks peak exhalation. The y-axis shows pupil size in Z scores. The horizontal zero line represents the average pupil size for each individual. Pupil size for females ($n = 59$ & 57 for nose & mouth breathing respectively) is depicted by the green circles connected by a solid line, and pupil size for males ($n = 30$ & 34 for nose & mouth breathing respectively) is depicted by the blue squares connected by a dashed line. The shaded areas in all line plots represent the 95% confidence intervals.

Please also include the number of participants in each protocol in the results section and figure captions and clarify why heart rate was monitored during the experiments.

Response: We have now included the number of participants in all the relevant sections. Heart rate was recorded as a control measure for the behavioral task, which is not the focus of the current manuscript. We have now added the following clarification to the methods section:

Heart rate data was collected as a control measure for the behavioral task of Experiment 3, which will not be included here.

The abstract section can be improved by including more detailed information about experiments (humans, number of participants), methods, experimental protocols, and main conclusions of the study.

Response: We rewrote the abstract and it now reads:

Respiration shapes brain activity and synchronizes sensory and exploratory motor actions, with some evidence suggesting it also affects pupil size. However, evidence for a coupling between respiration and pupil size remains scarce and inconclusive, hindered by small sample sizes and limited controls. Given the importance of pupil size in visual perception and as a reflection of brain state, understanding its relationship with respiration is essential. In five experiments using a pre-registered protocol—we systematically investigated how respiratory phase affects pupil size across different conditions.

In Experiment 1 ($n = 50$), we examined nasal and oral breathing at rest under dim lighting with nearby fixation points, then replicated these results under identical conditions in Experiment 2 ($n = 53$). Experiment 3 ($n = 112$) extended this to active visual tasks, while Experiment 4 ($n = 57$) extended this to controlled breathing at different paces under ambient lighting with distant fixation. Finally, in Experiment 5 ($n = 34$), individuals with isolated congenital anosmia (born without olfactory bulbs) were used as a lesion-type model during visual-auditory tasks to assess whether the respiratory-pupil link depends on olfactory bulb-driven oscillations.

Across all conditions—free and controlled breathing; different tasks, lighting, and fixation distances; and with and without olfactory bulb—we consistently found that pupil size is smallest around inhalation onset and largest during exhalation. We term this effect the Pupillary Respiratory-Phase (PRP) response, the fourth known mechanism influencing pupil size, alongside the pupillary light, near fixation, and psychosensory responses.

Lastly, while the speculation about connections between the preBötC and LC is intriguing, the references cited do not provide experimental evidence supporting the existence of noradrenergic projections from the preBötC to the LC, nor the presence of noradrenergic interneurons in the LC receiving preBötC inputs. Clarification and resolution of these issues are essential to ensure the study's conclusions are accurately represented and understood.

Response: We did not intend to claim that there are noradrenergic projections from the preBötC to the LC, but rather that the LC releases norepinephrine based on connections from the preBötC. We only write:

“In this context, the most prevalent manner in which respiration can affect pupil size is through excitatory inputs from the preBötC to the locus coeruleus, which are phase-locked to inspiration, leading to a respiration-regulated release of norepinephrine by the locus coeruleus (del Negro et al., 2018; Yackle et al., 2017) “.

The study by Yackle et al. 2017 (which we cite in this context) provides experimental evidence that Cdh9/Dbx1 neurons in the PreBötC directly project to and synapse on noradrenergic neurons in the contralateral LC, and via this connection provide excitatory input to the LC, and that Cdh9/Dbx1 preBötC neurons function as gateway neurons directly linking the preBötC to the locus coeruleus, and through it to the rest of the brain.

If we have misunderstood your comment, please let us know how.

Senior Editor's comments:

Comments to ensure the paper complies with the Statistics Policy:

A post-hoc statistical analysis should be used to confirm differences in pupil size during the respiratory cycle.

Response: The two-way repeated measures ANOVA confirmed that pupil size changes significantly over the course of the breathing cycle. Critically, the permutation testing we performed served as a post-hoc statistical analysis across the breathing cycle bins. The tests confirmed that during the vast majority of breathing bins, the pupil size is significantly different from a random distribution. We have now made this clearer in the manuscript. We write:

We employed permutation testing as a statistical post-hoc analysis to assess in which breathing bins pupil size was statistically different from what could be expected from a random distribution.

Comments to the Authors:

Thank you for submission of your research article to the Journal of Physiology for consideration. The article has been reviewed by experts in the field and found to require a major revision to address all of the concerns raised before a decision can be made for acceptability for publication. Please address all comments from the external referees and reviewing editor as well as addressing the list of requirements or publication in the journal including the statistical requirements.

Response: We have now revised the manuscript and believe to have addressed all the comments and fulfilled the journal requirements.

Please begin the Methods with the section Ethical Approval, rather than Participants. Begin with: Ethical approval was obtained from the national Swedish Ethical Review Authority (Dnr 2020-00972) and all participants signed informed consent prior to participation. The study conformed to the standards set by the Declaration of Helsinki, except for the lack of registration in a database before recruitment of the first participant (clause 35).

Response: We have made the suggested change.

Please incorporate all of the supplementary material into the manuscript. The Journal does not accept supplementary methods, results, figures and tables. There is no restriction on page length for the manuscript.

Response: We have now incorporated all the supplementary material into the manuscript.

Please revise the abstract to be more informative of design and outcomes.

Response: This has now been done (see response above for more details).

A post-hoc statistical analysis should be used to confirm differences in pupil size during the respiratory cycle.

Response: This was already conducted via the permutation testing. See the response above for more details.

Reviewers' comments

Reviewer 1 comments:

Respiratory modulation of pupil diameter has been known for decades. However, few direct studies on this phenomenon in humans are available, and published observations suffer from methodological limitations. Schaefer et al. aim to fill this gap in knowledge by directly studying how pupil diameter changes with breathing phase in humans at rest and during a visual task and when subjects are asked to breathe through their nose or mouth. The authors find that pupil size is maximal during peak exhalation and minimal during inhalation onset. Further, this pattern of respiratory modulation of pupil size, which they term the Respiratory-Pupillary Phase Effect (RPPE), does not differ whether subjects are breathing through their nose or mouth, or whether subjects are at rest or engaged in a visual perception task.

The manuscript presents straightforward studies that rigorously characterize the effects of breathing on pupil size in humans. The larger sample size, rigorous study design, and smaller phase bins are significant improvements over previous studies. The results are high quality and are an important contribution to our understanding of these basic physiological processes, particularly because the studies are conducted in humans. The lack of effect of breathing route and the effect of engagement in a task on RPPE are also novel contributions to the field. While the manuscript is generally clear and the conclusions are supported by the data, a number of issues should be addressed that would improve the manuscript.

1) The authors present a key, erroneous assumption about the prevailing view as being that pupil size is maximal during inhalation and minimal during exhalation. While I was unable to access the pre-Borgdorff 1975 references, I do not find this explicit claim in Borgdorff or the other more recent cited references. While one may draw this inference from the figures in Borgdorff, citing specific publications that make this claim would strengthen this argument. It appears that the prevailing view described in many of these studies is that pupil size changes with breathing phase and, perhaps more specifically, that pupil dilates during inhalation and constricts with exhalation, which is consistent with Borgdorff and the results of this study. In my view, the significance of this manuscript would not be significantly diminished if statements related to the minimum/maximum claim in the key points, abstract, introduction, and discussion were tempered or removed, especially if this claim is not explicitly made in published literature.

Response: The reviewer is correct that Borgdorff does not explicitly state that pupil size is largest during inhalation, but from our discussions with others in the field, this is the key takeaway many derive from the figures. Importantly, we show that, unlike Borgdorff's findings, pupil dilation and constriction occur during both phases of the breathing cycle and are not confined to one phase or the other. We have removed the original framing in the introduction, and tempered it throughout the manuscript, but we still highlight Borgdorff's findings and their implications in relation to our study in the discussion. This is crucial, as Borgdorff's cat paper remains the main paper in the field that many researchers reference.

We now write in the discussion:

Notably, dilation and constriction occur during both phases of the breathing cycle, but not to the same extent. Specifically, the pupils dilate through most of inhalation and the early phase of exhalation, whereas pupil constriction primarily happens during the latter part of exhalation. This contrasts partially with previous assumptions, primarily based on Borgdorff's (1975) landmark study on cats, where pupil dilation was observed during inhalation and constriction during exhalation in lightly anesthetized or tranquilized cats. However, this effect disappeared in fully awake or heavily anesthetized cats. Despite Borgdorff (1975) acknowledging this limitation, his findings have been interpreted as evidence of a

broader physiological mechanism. Additionally, although not explicitly stated, the figures in Borgdorff's (1975) paper depicted maximal dilation during inhalation and minimal dilation during exhalation, suggesting that pupils are largest during inhalation and smallest during exhalation. However, here we show conclusively that this is not the case, and that in fact the effect is in the opposite direction.

2) While combining data from Experiments 1 and 2 is justified, the statistical descriptions that are currently in Supplementary Materials for the individual datasets should be incorporated into the Results of the main text to support grouping the datasets. Supplementary Figures and Tables can remain in Supplementary Materials, but should be individually/specifically referenced in the text. There are also sentences related to one-way repeated measures ANOVAs in Supplementary Materials, where it is unclear whether these are for relative or absolute pupil size and what the rationale is for this statistical comparison. These should be more clearly described and moved to the main Results as well.

Response: As the journal policy does not allow us to have supplementary materials, we have now put the results of Experiment 1 and 2 in the main text and have instead decided to remove the combined results to make the manuscript not too cluttered. Additionally, we have clarified both the reasoning for the one-way repeated measures ANOVAs (see below), as well as on which data they were performed.

To make use of all the recordings (and not just the ones of the participants that had complete recordings for both nose and mouth breathing) we also conducted separate one-way ANOVAs for nose and for mouth breathing.

3) It would be helpful to have a description of how the circular mean direction of individual pupil size is calculated in the Methods. In the results, is the "average mean vector direction" the same as the "circular mean direction"? Is the magnitude of the vector ignored? Is the minimal pupil size 180 degrees from the angle of the circular mean direction?

Response: The `circ_mean.m` function calculates the mean direction for circular data. The mean direction does not necessarily correspond to the phase bin where pupil size is largest, and neither is the minimal pupil size necessarily 180 degrees opposed to the mean direction. No vector magnitude is calculated. We have now added the following to the methods section as clarification:

The `circ_mean.m` function uses the following formula to calculate the mean direction for circular data:

$$direction = angle(sum(weights * (i * radians)^2))$$

Where the *weights* correspond to the average normalized pupil size for each breathing bin, and the *radians* correspond to the angle of each breathing bin in radians. The *angle* function returns the phase angle, in radians, of a complex number.

4) In the Discussion, it is mentioned that the changes in pupil size were "highly significant yet relatively small." Is this statement based on normalized z-score or absolute measurements of pupil size (in mm)? Some description of effects sizes and/or absolute measurements in Results might help justify this statement.

Response: Upon reflection, we believe the original statement lacked nuance and could be misleading. In Experiment 4, we used an updated version of the eye-tracking software that provides a much more

precise estimate of pupil size in millimeters rather than pixels. This allows for a clearer interpretation of the effect size and situates it within the broader context of the maximum dilation/constriction range of the human pupil.

We have now elaborated on this point in more detail in the discussion section.

The observed changes in pupil size for the PRP response had an effect size ranging from 0.11 to 0.23 mm change on average across the respiratory cycle depending on the breathing condition (see Experiment 4), or ranging from 0.12 to 0.35 Z-scores across experiments (see Experiments 1-5). Considering that the human pupil can vary by up to ca 6 mm from its smallest (~2 mm) to largest (~8 mm) size under normal circumstances (Mathôt, 2018), this observed effect represents about 1.8-3.8% of the total range. This effect is too small to be readily apparent in individual breathing cycles, but is readily apparent when averaging pupil size over several breathing cycles. One way to contextualize this effect size is to compare it to the three other known pupil responses (Mathôt, 2018). The pupillary light response, which constricts the pupil in response to light, and the pupillary near response, which constricts the pupil during near fixation, can both change the pupil up to several millimeters. In contrast, the pupillary psychosensory response, which dilates the pupil in response to cognitive activity (e.g., arousal and mental effort), results in much smaller changes, generally only a few tenths of a millimeter. The PRP response is comparable in effect size to the pupillary psychosensory response. However, the PRP response differs from the psychosensory response in its cyclic nature, involving both dilation and constriction, making it the only known pupil response with this characteristic.

5) I was confused by the data related to the changes in absolute pupil size (Figures S4 and S7). The 2x2 repeated measures ANOVA describes time as a variable, and the figure shows 0 to 300 s on the x-axis, which would correspond to the duration of the recording. However, the figure legend mentions phase bins, which would not make sense for breathing on this time scale. Either the data show a steady decrease in absolute pupil size over the 5 minute duration of the task or a maximum pupil size at the onset of inhalation (if x-axis is phase), which is inverted from the normalized data. Please clarify whether the data is showing absolute pupil size over the duration of the task or over cycle phase and discuss the results in the Discussion.

Response: Thank you for noticing this. The figure legend was wrong, it depicts change over time, not over phase bins. We have now corrected this mistake.

6) A minority of subjects appeared to have a different phase relationship between breathing and pupil size, i.e., maximum pupil size during inhalation, that may have changed when engaged in a visual task. If possible, this source of intersubject variability and the effect of engagement in a visual task should be analyzed and discussed.

Response: We believe that the main difference between the pupil size at rest and the pupil size during the visual task is the recording length. There is more variation in the measurements at rest because the recording time is almost an order of magnitude shorter. With more data, the variance in the pupil size dynamics is averaged out and the PRP response becomes clearer.

As a crude analysis we checked the 10 subjects which had their mean vector direction point towards inhalation for both nose and mouth breathing at rest (Experiment 1 and 2). None of these subjects had their mean vector direction point towards inhalation for both nose and mouth breathing during the visual task (Experiment 3). This strengthens our belief that this variability in the phase relationship

between breathing and pupil size has not so much to do with the subjects per se, as with the PRP response not being very strong.

When analyzing only a 5-minute subset of the 45-minute visual task data (i.e., similar to the rest condition), there is also more variability in the results, similar to the data recorded at rest. Thus, we believe that these variations mainly stem from shorter recording time and what would be expected from stochastic measurement error.

We agree that it would be of interest to further study what influences the PRP response and causes variability between subjects and breathing cycles. However, we believe that to be outside the scope of this current manuscript as our paradigms were not designed to capture the different sources shaping individual variability (e.g., genetic background, and other relevant background information that we do not have access to).

7) In the paragraphs discussing possible mechanisms underlying PRPE (paragraphs 4 and 7 in Discussion), the multisynaptic, neuromodulatory pathways may be sources of latency that introduce a phase delay or alter the phase relationship between neural activity in preBötzinger Complex and pupil dilation. This possibility should be discussed.

Response: Yes, there might be phase delays at different parts of the internal system. Also, it should be noted that there might be other lags to consider as well. Our measurement of the respiratory phase for example is based on the observed change in air pressure within the cannula, which, by nature, occurs after the initiation of neuronal activity in the preBötzinger Complex. To the best of our knowledge, the precise phase delay between this neuronal activation and the corresponding measurable changes in air pressure is unknown. For example, one could easily imagine a scenario where the lag between the preBötzinger Complex and pupil size change differs from the lag between the preBötzinger Complex and the change in air pressure. However, as we do not have any hypothesis about different phase lags internal or external, this would be pure speculation, and we would prefer not to elaborate on this in the discussion and keep the text as close to our data as possible.

8) “Negro et al., 2018” should be “del Negro et al., 2018”.

Response: We corrected this reference.

Reviewer 2 comments:

In the present study, the authors examined relative pupil size during respiration during both rest and while actively engaged in a visual attention experiment. They found that pupil size tended to increase as people inhaled peaked around maximum inhalation, then pupil size decreased with exhalation.

I have only a few minor questions I would like to see addressed.

1) I am curious about several findings mentioned. And while I do not believe additional data are necessary, I believe they would be informative. For example, the authors mention a possible difference between controlled and passive breathing. This seems like an easy follow-up experiment to perform. I

could imagine a set of competing hypotheses, too. One hypothesis would be that self-regulated, controlled breathing would amplify the effect of respiration on pupil size because it would recruit attention, and perhaps consequently, the LC-NE system. Another hypothesis is that passive, uncontrolled breathing is regulated by the brainstem (which houses the LC), and therefore controlling one's breathing might wrest control over respiration away from the brainstem. But I don't know enough about the physiology of the connection between the brain and the muscles of the lungs to know if that's plausible. I would therefore be curious to see a comparison of controlled vs. passive breathing. Otherwise, is there extant data on controlled vs. passive breathing on pupil size in people?

Response:

We appreciate the constructive comments. To this end we conducted a controlled breathing experiment. Here we asked people to conduct three blocks of nasal respiration. Two of these were controlled breathing, one at 8 breaths/minute, and the other at 16 breaths/minute, which is respectively slower and faster than the average normal breathing pace. The third was a control task, with free, uncontrolled breathing. In accordance with our other four experiments the PRP response did not change significantly as a function of controlled breathing or pace.

Regarding the theoretical assumptions of controlled breathing, we have expanded on this in the manuscript. Specifically, while normal, automatic breathing is regulated by the preBötzinger Complex (preBötzc) in the medulla in response to input from mechanoreceptors and chemoreceptors, during active breathing, control of the respiratory centers in the medulla shifts to the primary motor region of the cerebral cortex. Additionally, during active expiration, expiratory pump muscles, particularly the internal intercostal and abdominal muscles, are engaged—this does not occur during passive expiration (De Troyer et al., 2005; Mortola, 2013; Welch et al., 2019). All these changes are potential mechanisms by which the observed response could have been modified. However, as we show, the PRP response is robust under these conditions as well, indicating that it is likely shaped by early brainstem circuits.

We now write:

During autonomous, passive breathing, the rhythm and motor pattern of respiration is generated by the respiratory centers in the pons and the medulla guided by the information relayed from chemo- and mechanosensors. In contrast, during controlled, volitional breathing, respiration is modulated by the cortex (for review see Trevizan-Baú et al., 2024). Additionally, during active expiration, expiratory pump muscles—particularly the internal intercostal and abdominal muscles—are engaged, which does not occur during passive expiration (De Troyer et al., 2005; Mortola, 2013; Welch et al., 2019). These changes represent potential neural mechanisms by which the observed pupil response could have been modified. However, as demonstrated in Experiment 4, neither active control over inhalation and exhalation, nor the pace of breathing diminish the PRP response, although these factors might influence its magnitude or modify its shape (see Experiments 1-4).

2) Another thing that struck my interest was the potential for regulation of the LC-NE system by respiration. Many meditation techniques, some of which have been successfully employed as interventions to improve attention, use controlled and focused breathing. I wondered whether one potential mechanism is a momentary, or perhaps lasting, regulation of the LC-NE system, which might

have downstream benefits on attention. That is, can breathing be a way to exert "control" over the LC-NE system? Is there any existing data that might speak to this?

Response: Yes, there is some research showing that breathing can influence LC activity. Yackle et al. (2017) showed that a neuronal subpopulation in the mouse preBötC positively regulate noradrenergic neurons in the LC. They hypothesized that this coordinates the animal's arousal state with its breathing pattern, with slow and regular breathing leading to a calm and relaxed state, whereas rapid or irregular breathing would promote or maintain higher levels of arousal (Yackle et al., 2017). Furthermore, Melnychuk et al. (2018) present evidence of synchronization between respiration and LC activity in humans. They hypothesize that this provides a link between respiration and attention via the LC, and that this coupling can be influenced by breath-focused practices (Melnychuk et al., 2018).

Otherwise, I found this to be a very well conducted set of experiments and a well written manuscript. This was a topic about which I knew almost nothing and it seems important that the results, which were robust and clear, violate the traditional wisdom.

References

- De Troyer, A., Kirkwood, P. A., & Wilson, T. A. (2005). Respiratory Action of the Intercostal Muscles. *Physiological Reviews*, 85(2), 717–756. <https://doi.org/10.1152/physrev.00007.2004>
- Mathôt, S. (2018). Pupillometry: Psychology, physiology, and function. *Journal of Cognition*, 1(1).
- Melnychuk, M. C., Dockree, P. M., O'Connell, R. G., Murphy, P. R., Balsters, J. H., & Robertson, I. H. (2018). Coupling of respiration and attention via the locus coeruleus: Effects of meditation and pranayama. *Psychophysiology*, 55(9), e13091. <https://doi.org/10.1111/psyp.13091>
- Mortola, J. P. (2013). Lung Viscoelasticity: Implications on Breathing and Forced Expiration. *Clinical Pulmonary Medicine*, 20(3), 144. <https://doi.org/10.1097/CPM.0b013e31828fc9d6>
- Trevizan-Baú, P., Stanić, D., Furuya, W. I., Dhingra, R. R., & Dutschmann, M. (2024). Neuroanatomical frameworks for volitional control of breathing and orofacial behaviors. *Respiratory Physiology & Neurobiology*, 323, 104227. <https://doi.org/10.1016/j.resp.2024.104227>
- Welch, J. F., Kipp, S., & Sheel, A. W. (2019). Respiratory muscles during exercise: Mechanics, energetics, and fatigue. *Current Opinion in Physiology*, 10, 102–109. <https://doi.org/10.1016/j.cophys.2019.04.023>
- Yackle, K., Schwarz, L. A., Kam, K., Sorokin, J. M., Huguenard, J. R., Feldman, J. L., Luo, L., & Krasnow, M. A. (2017). Breathing control center neurons that promote arousal in mice. *Science*, 355(6332), 1411–1415. <https://doi.org/10.1126/science.aai7984>

Dear Dr Schaefer,

Re: JP-RP-2024-287205R1 "The Pupillary Respiratory-Phase Response: Pupil size is smallest around inhalation onset and largest during exhalation" by Martin Schaefer, Sebastiaan Mathot, Mikael Lundqvist, Johan N Lundstrom, and Artin Arshamian

Thank you for submitting your manuscript to The Journal of Physiology. It has been assessed by a Reviewing Editor and by 3 expert referees and we are pleased to tell you that it is acceptable for publication following satisfactory revision.

REVISION CHECKLIST:

We look forward to receiving your revised submission.

Yours sincerely,

Harold Schultz
Senior Editor
The Journal of Physiology

Reviewing Editor's comments:

I appreciate the authors for addressing all the comments and making the necessary adjustments to the manuscript, including new experiments. Both referees acknowledged that the revised version showed improvement; however, further comments regarding data analyses arose after assessing this version, which require the authors' attention. Additionally, I thank the authors for responding to my comments, particularly the one about clarifying the possible involvement of the preBötzing complex and locus coeruleus connections. However, part of the text in the response to the referee document ("leading to a respiration-regulated release of norepinephrine by the locus coeruleus") differs from the text in the discussion in the revised manuscript ("leading to a respiration-regulated release of norepinephrine in the locus coeruleus"). Please correct the text in the manuscript to prevent misinterpretation. Moreover, the last three paragraphs of the Introduction section provide details about the experiments and conclusions. As per the JPhysiol guidelines, I recommend that the authors relocate this information to other relevant sections, if necessary, and instead present the study's rationale.

Senior Editor:

Comments to ensure the paper complies with the Statistics Policy:

Please see comments from the statistics editor [Referee 3]

Comments to the authors:

Thank you for submission of your revised research article to the Journal of Physiology for consideration. The article has been reviewed by the original referees and the Journal's statistics editor and found to require additional revision to address remaining concerns raised. Please address all comments from the external referees and Journal editors as well as confirming the list of requirements or publication in the journal.

Referee #1:

Please see attached file [JP-RP-2024-287205R1_Referee 1 Review Attachment 1.pdf]

Referee #2:

I appreciate the authors addition of a controlled breathing experiment based on my question. I also appreciate the citations connecting controlled breathing exercises to the LC and to attention.

Referee #3:

The data analysis utilises advanced statistical methods well-suited to the dataset. However, the analysis would benefit from enhanced clarity, more detailed explanations, and explicit procedural steps to improve reproducibility and overall scientific rigour.

While the authors employed permutation testing as a post-hoc statistical analysis to identify breathing bins in which pupil size significantly deviated from random distributions, they did not apply corrections for multiple comparisons. Conducting 18 individual tests without adjustment increases the likelihood of a false positive result to approximately 60%. To address this, correction methods such as the Bonferroni adjustment, Benjamini-Hochberg procedure, or permutation-based family-wise

error rate corrections should be considered and explicitly reported.

The use of repeated measures ANOVAs is appropriate; however, the manuscript does not specify whether key assumptions, such as sphericity, were tested. If sphericity was violated, corrections such as the Greenhouse-Geisser adjustment should have been applied, and these adjustments need to be reported. Furthermore, the assumption of independence in ANOVA requires careful attention to avoid pseudo-replication, where each row of data should represent a distinct participant. If nested or repeated measures data structures are present, linear mixed models incorporating participant IDs as random effects may be a more suitable analytical approach. The authors are encouraged to clarify how they have addressed potential pseudo-replication issues and ensured the independence of their data within the ANOVA framework.

END OF COMMENTS

In this revised manuscript, Schaefer et al. significantly reorganized and rewrote their manuscript and added intriguing experiments related to volitional breathing and the role of the olfactory bulb in the pupillary respiratory-phase response in patients that were born without this structure. The changes addressing the issues raised in the initial review are largely satisfactory and the additional experiments are novel and enhance the potential impact of their results. An issue related to the post-hoc permutation testing remains and addressing several minor issues would improve the clarity of the manuscript.

1) Post-hoc permutation test – The authors specify in the revision that they use the permutation approach for statistical post-hoc analysis of their data. However, the revised description of how their statistics are conducted and several of their statistically significant results are confusing, and further clarification and/or justification for their statistical approaches would be helpful.

a) The parametric ANOVA is generally followed by a post-hoc test (Tukey, t-test with Bonferroni-Holm correct, etc.) to determine which groups are significantly different from one another. The permutation test described seems different from a pairwise permutation test and, instead, appears to test whether the individual bins are significantly different from a random/shuffled distribution, which does not seem to be related to the ANOVA. Thus, it is still unclear which bins are statistically different from one another and why a parametric post-hoc test could not be applied.

b) The authors appear to be using the permutation approach to test the null hypothesis that the mean of each bin is different from chance. However, since they draw their random sample across all phases, I am not certain this approach is valid for a bin by bin comparison since the bins are not independent (and why a multiple comparison correction is generally recommended). It seems from intuition and the central limit theorem that the permutation approach would produce a resampled distribution where the z-score was 0 at every phase bin, i.e., no phase modulation, and that there should be bins in the actual data that are not significantly different from the subject's mean pupil size when the pupil size crosses from its maxima and minima. What is surprising were the results in Experiments 3 and 5 where every phase bin was significantly different, which raises questions about how this approach is being applied. While inferences might be drawn about whether the observed distribution across the breathing cycle is different from a random distribution, it seems to me that an approach that permuted observations within bins would be needed to be able to determine which bins are significantly different from "random" data.

c) The description of how they are populating their random distributions for permutation testing could be made clearer. The authors state they are using randomly selected pupil size observations and repeating the selection 10,000 times. It is unclear whether they are drawing from the 18 averaged pupil size values for each of the 30-100 subjects in each experiment or drawing from the ~30,000-880,000 observations and then calculating a mean for each bin. For the latter, my understanding is that test statistics should be generated for all permutations of the data, so it seems that generating 10,000 distributions is insufficient to produce a distribution of the test statistic and calculate the proportion of permutations that are smaller/larger than the observed dataset.

Minor issues

1) For each “Experiment” section in the Results, it would help readability to have a brief description of the rationale/experimental conditions at the beginning prior to reporting the breathing measurements.

2) For each “Experiment” section in the Results, the range of absolute pupil size change (in mm) that correspond to the z-scores should be reported in the Results even if statistics are not performed on these measurements.

3) The change in absolute pupil size over time is reported for Experiments 1-3; however, these data are not discussed or placed in any context in the text besides being a preregistered measure. The rationale for reporting these statistically significant results and their interpretation should be included.

4) Pupil size derivative calculation and interpretation – How the pupil size derivative was calculated should be described in the Methods. While not necessary, statistical analysis of the derivative, similar to that performed on the normalized pupil size, would help strengthen their conclusion about constriction and dilation occurring during both inhalation and exhalation. I understand that this statement is qualified in the text, but the generally very small negative values (and positive values in some experiments) at the beginning of inhalation may not be statistically significant or physiologically relevant. In the absence of statistics, the statement at the beginning of Results in the General section (“Pupil dilation occurred predominantly throughout inhalation...”, p. 15) seems to me more accurate than repeatedly saying constriction and dilation occur during both inhalation and exhalation.

5) Phase relationship between neural activity and pupil size –The authors respond that they wish to avoid speculating about phase shifts from neural activity to the pupil measurement in their discussion of proposed neural mechanisms. However, in the absence of a discussion on lags, maximal pupil dilation during exhalation is inconsistent with a preBötzC driven mechanism, since this region is only active during inspiration. If they wish to propose the very plausible model that preBötzC activity is driving pupil dilation via locus coeruleus, then how inspiratory activity from preBötzC leads to maximal dilation during a phase where preBötzC is largely silent should be reconciled, if even briefly. Additionally, sources of phase shifts may depend on experimental setup, e.g., how breathing phase is measured, or species, e.g., between cats and humans, that may explain the inconsistencies between this study and Borgdorff and others.

13 January, 2025

Harold Schultz

Senior Editor

The Journal of Physiology

=====

Dear Prof. Schultz,

Attached is the second revision of our manuscript for consideration for publication in *The Journal of Physiology*, with the title "The Pupillary Respiratory-Phase Response: Pupil size is smallest around inhalation onset and largest during exhalation" (Re: JP-RP-2024-287205R1). We appreciate the time and effort of the editors and referees that went into reviewing our manuscript and giving us feedback.

We have now revised our manuscript based on the points raised, and believe this has again improved the work.

The main changes to the manuscript consist of sphericity checks and corrections applied to our ANOVAs, as well as a change in our permutation testing approach to include permutation-based family-wise error rate corrections. This has affected some of our p -values and strengthened the statistical robustness of our results, without changing any of our conclusions.

Further changes include changes in phrasing and clarifications which should improve the readability of the manuscript.

The following is a point-by-point response to the reviewers' comments. Our responses are marked in blue, and the changes in the manuscript are highlighted in yellow. We hope this revised submission will meet the reviewers' expectations and be suitable for publication in *The Journal of Physiology*.

On behalf of all the authors,

Martin Schaefer

=====

Reviewing Editor's comments:

I appreciate the authors for addressing all the comments and making the necessary adjustments to the manuscript, including new experiments. Both referees acknowledged that the revised version showed improvement; however, further comments regarding data analyses arose after assessing this version, which require the authors' attention. Additionally, I thank the authors for responding to my comments, particularly the one about clarifying the possible involvement of the preBötzing complex and locus coeruleus connections. However, part of the text in the response to the referee document ("leading to a respiration-regulated release of norepinephrine by the locus coeruleus") differs from the text in the discussion in the revised manuscript ("leading to a respiration-regulated release of norepinephrine in the locus coeruleus"). Please correct the text in the manuscript to prevent misinterpretation.

We have now corrected this oversight.

Moreover, the last three paragraphs of the Introduction section provide details about the experiments and conclusions. As per the JPhysiol guidelines, I recommend that the authors relocate this information to other relevant sections, if necessary, and instead present the study's rationale.

We slightly modified one of the paragraphs, and removed the last two paragraphs from the introduction. Instead we added some additional experimental descriptions and rationale to the results section for each experiment, in line with the feedback from Referee # 1.

Senior Editor's comments:

Comments to ensure the paper complies with the Statistics Policy:

Please see comments from the statistics editor [Referee 3]

Comments to the authors:

Thank you for submission of your revised research article to the Journal of Physiology for consideration. The article has been reviewed by the original referees and the Journal's statistics editor and found to require additional revision to address remaining concerns raised. Please address all comments from the external referees and Journal editors as well as confirming the list of requirements or publication in the journal.

Referee #1 comments:

Please see attached file [JP-RP-2024-287205R1_Referee 1 Review Attachment 1.pdf]

In this revised manuscript, Schaefer et al. significantly reorganized and rewrote their manuscript and added intriguing experiments related to volitional breathing and the role of the olfactory bulb in the pupillary respiratory-phase response in patients that were born without this structure. The changes addressing the issues raised in the initial review are largely satisfactory and the additional experiments are novel and enhance the potential impact of their results. An issue related to the post-hoc permutation testing remains and addressing several minor issues would improve the clarity of the manuscript.

1) Post-hoc permutation test – The authors specify in the revision that they use the permutation approach for statistical post-hoc analysis of their data. However, the revised description of how their statistics are conducted and several of their statistically significant results are confusing, and further clarification and/or justification for their statistical approaches would be helpful.

a) The parametric ANOVA is generally followed by a post-hoc test (Tukey, t-test with Bonferroni-Holm correct, etc.) to determine which groups are significantly different from one another. The permutation test described seems different from a pairwise permutation test and, instead, appears to test whether the individual bins are significantly different from a random/shuffled distribution, which does not seem to be related to the ANOVA. Thus, it is still unclear which bins are statistically different from one another and why a parametric post-hoc test could not be applied.

In this paper, our primary focus is on if and how pupil size varies over the breathing cycle. With the ANOVA we determined that pupil size does vary over the breathing cycle, and with the permutation test we identifying which bins, are significantly different from zero/shuffled distribution. Determining whether bins differ from each other is a separate question, for which we have no specific hypothesis.

b) The authors appear to be using the permutation approach to test the null hypothesis that the mean of each bin is different from chance. However, since they draw their random sample across all phases, I am not certain this approach is valid for a bin by bin comparison since the bins are not independent (and why a multiple comparison correction is generally recommended). It seems from intuition and the central limit theorem that the permutation approach would produce a resampled distribution where the z-score was 0 at every phase bin, i.e., no phase modulation, and that there should be bins in the actual data that are not significantly different from the subject's mean pupil size when the pupil size crosses from its maxima and minima. What is surprising were the results in Experiments 3 and 5 where every phase bin was significantly different, which raises questions about how this approach is being applied. While inferences might be drawn about whether the observed distribution across

the breathing cycle is different from a random distribution, it seems to me that an approach that permuted observations within bins would be needed to be able to determine which bins are significantly different from “random” data.

Thank you for this suggestion. Based on the feedback from Referees 1 and 3, we have revised our permutation test to include a permutation-based family-wise error rate correction. We now generate a null distribution for each permutation separately for the data of each phase bin. Below is the updated description of the new permutation test from the methods section:

To determine whether the mean normalized pupil size for each of the 18 breathing bins differed significantly from zero, we conducted a nonparametric permutation test. This approach, which deviated from our pre-registered permutation test, was adopted based on reviewer feedback to avoid parametric assumptions about the data distribution and to provide robust control of the family-wise error rate (FWER) across multiple comparisons.

For each breathing bin, we computed the one-sample t-statistic as:

$$t = \frac{\bar{x}}{s/\sqrt{n}}$$

where \bar{x} is the mean normalized pupil size, s is the standard deviation, and n is the number of observations in the breathing bin.

The null hypothesis assumed that the mean normalized pupil size for each breathing bin was zero. To generate the null distribution, we performed 10,000 permutations for each breathing bin. During each permutation, the signs of the observations within the breathing bin were randomly flipped, preserving the magnitude and variability of the original data while simulating a dataset consistent with the null hypothesis.

To account for multiple comparisons across the 18 breathing bins, we used a maximum-statistic approach. For each permutation, we computed the t-statistics for all breathing bins and recorded the maximum absolute t-statistic across breathing bins. This procedure generated a null distribution of maximum test statistics, which was used to control the FWER.

For each breathing bin, the observed t-statistic was compared to the null distribution of maximum t-statistics. The family-wise error-corrected p -value for each breathing bin was

calculated as the proportion of permuted maximum t-statistics greater than or equal to the observed absolute t-statistic.

c) The description of how they are populating their random distributions for permutation testing could be made clearer. The authors state they are using randomly selected pupil size observations and repeating the selection 10,000 times. It is unclear whether they are drawing from the 18 averaged pupil size values for each of the 30-100 subjects in each experiment or drawing from the ~30,000-880,000 observations and then calculating a mean for each bin. For the latter, my understanding is that test statistics should be generated for all permutations of the data, so it seems that generating 10,000 distributions is insufficient to produce a distribution of the test statistic and calculate the proportion of permutations that are smaller/larger than the observed dataset.

Our new permutation testing approach resolves this issue.

Minor issues

1) For each “Experiment” section in the Results, it would help readability to have a brief description of the rationale/experimental conditions at the beginning prior to reporting the breathing measurements.

We have added a short description to each Experiment in the Results section. We now write for each of the Experiments respectively:

In Experiment 1, we measured nasal and oral respiration and pupil size in participants under dim conditions at rest, with a nearby fixation point, and upright head position. This allowed us to determine the potential relationship between respiratory cycles and pupillary dynamics under simple and controlled conditions.

To ensure the robustness of the findings from Experiment 1, which was exploratory, we performed a direct replication in Experiment 2 in an independent group of participants, but while keeping the experimental conditions the same.

In Experiment 3, we determined whether results from Experiments 1 and 2, in which participants did not perform any task, persisted during active visual perception under otherwise the same conditions. So we measured breathing and pupil size while participants performed a 3-alternative forced choice visual detection task.

In Experiment 4, we tested if the results from the previous experiments also persisted during controlled breathing (fast and slow) under more brightly lit conditions and with a distant point of focus. Because controlled breathing is known to activate different brain regions than

passive breathing, and light intensity and fixation distance are known to influence pupil size, these factors have the potential to influence the effect that breathing phase has on pupil size (for review see (Mathot, 2018; Trevizan-Baú et al., 2024)).

Finally, in Experiment 5, we tested whether the effect of respiration phase on pupil size persisted in the absence of the olfactory bulb respiratory oscillator during nasal respiration. Since the olfactory bulb is known to generate important global respiratory-coupled oscillations, its absence might influence the effect of respiration phase on pupil size (Karalis & Sirota, 2022; Kay et al., 2009; Tort et al., 2018; Zelano et al., 2016). To investigate this, we used a lesion-type model involving participants born without olfactory bulbs, a rare disorder causing isolated congenital anosmia, but otherwise normal health, alongside control participants. Both groups were tested during visual and auditory tasks while lying down.

2) For each “Experiment” section in the Results, the range of absolute pupil size change (in mm) that correspond to the z-scores should be reported in the Results even if statistics are not performed on these measurements.

We have only reported this for Experiment 4, as we only have accurate absolute pupil size measures for this experiment. We mention this limitation in the methods section and prefer not to calculate absolute pupil size ranges for the other Experiments as we know (in hindsight) that those values are not reliable.

3) The change in absolute pupil size over time is reported for Experiments 1-3; however, these data are not discussed or placed in any context in the text besides being a preregistered measure. The rationale for reporting these statistically significant results and their interpretation should be included.

We have now added the following elaboration to the methods section:

Since we also had the hypothesis that pupil size might be larger during mouth than nose breathing, and that this effect might only establish itself over time, we conducted an additional two-way (time x breathing route) repeated measures ANOVA for Experiments 1 and 2. For this, we divided the 5-minute pupil size recordings into 18 evenly spaced time bins, and compared the average absolute pupil size for each time bin and each breathing route.

Furthermore, we added the following line to the results section of Experiment 1 under Absolute pupil size:

The decrease in pupil size over time might indicate a decrease in arousal as participants become more relaxed over the course of the recording.

And similarly added the following to the results section of Experiment 2:

with pupil size decreasing over the course of the recording

And added a sentence in the discussion:

Also in absolute pupil size measures, we showed that pupil size is not significantly affected by breathing route over time (Experiment 1 and 2).

4) Pupil size derivative calculation and interpretation – How the pupil size derivative was calculated should be described in the Methods. While not necessary, statistical analysis of the derivative, similar to that performed on the normalized pupil size, would help strengthen their conclusion about constriction and dilation occurring during both inhalation and exhalation. I understand that this statement is qualified in the text, but the generally very small negative values (and positive values in some experiments) at the beginning of inhalation may not be statistically significant or physiologically relevant. In the absence of statistics, the statement at the beginning of Results in the General section (“Pupil dilation occurred predominantly throughout inhalation...”, p. 15) seems to me more accurate than repeatedly saying constriction and dilation occur during both inhalation and exhalation.

We have now added the following paragraph to the methods section to describe the pupil size derivative calculation:

To examine how pupil size changes throughout the breathing cycle, we calculated the pupil size derivative for each phase bin. First, we determined the change in pupil size between consecutive observations across the entire recording. Next, for each breathing cycle, we summed these changes within each phase bin, resulting in a value representing pupil size change for each bin in that cycle. Finally, we averaged these values across all breathing cycles to obtain the pupil size derivative for each phase bin.

Furthermore, we have changed the wording regarding pupil dilation and constriction occurring during both inhalation and exhalation to better reflect what can be inferred from the figures. We now write the following:

Results section Experiment 1

To highlight the difference between pupil size and changes in pupil size, we also plotted the change in pupil size over the course of the breathing cycle (see Figure 2C). This plot revealed that while the majority of pupil size dilation occurred during inhalation and the majority of pupil constriction occurred during exhalation, the two processes were not strictly isolated to inhalation and exhalation respectively.

Results section Experiment 2

Similar to Experiment 1, during Experiment 2, the majority of pupil size dilation occurred during inhalation and the majority of pupil constriction occurred during exhalation (see Figure 2C).

Results section Experiment 3

Furthermore, during the visual task the majority of inhalation consisted of pupil size dilation, and pupil constriction predominantly occurred during exhalation (see Figure 3C).

Results section Experiment 4

Furthermore, we again found that the majority of pupil size dilation occurred during inhalation and the majority of pupil constriction occurred during exhalation (see Figure 4C). However, the pattern of pupil dilation and constriction differed slightly for the paced breathing condition as compared to the normal breathing.

Results section Experiment 5

Assessing the change in pupil size revealed that most of inhalation consisted of pupil dilation and most of pupil constriction occurred during the second half of exhalation for both groups (see Figure 5C).

Discussion

Notably, while inhalation is predominantly marked by pupil size dilation, and the majority of pupil constriction occurs during exhalation, the two processes do not seem to be strictly isolated to inhalation and exhalation respectively. This adds some nuance to previous assumptions, primarily based on Borgdorff's (1975) landmark study on cats, where pupil dilation was observed during inhalation and constriction during exhalation in lightly anesthetized or tranquilized cats.

5) Phase relationship between neural activity and pupil size –The authors respond that they wish to avoid speculating about phase shifts from neural activity to the pupil measurement in their discussion of proposed neural mechanisms. However, in the absence of a discussion on lags, maximal pupil dilation during exhalation is inconsistent with a preBötzC driven mechanism, since this region is only active during inspiration. If they wish to propose the very plausible model that preBötzC activity is driving pupil dilation via locus coeruleus, then how inspiratory activity from preBötzC leads to maximal dilation during a phase where preBötzC is largely silent should be reconciled, if even briefly. Additionally, sources of phase shifts may depend on experimental setup, e.g., how breathing phase is measured, or species, e.g., between cats and humans, that may explain the inconsistencies between this study and Borgdorff and others.

We added the following sentences to the discussion to address the phase shift between PreBötzc activity and maximum pupil dilation:

Even though we believe the preBötzc to be the driver of this effect, maximum pupil size (~mid exhalation) does not occur simultaneously with maximum preBötzc activity (~inhalation onset). This indicates that it takes some time from the firing of the preBötzc to the maximal pupil dilation by the constriction of the iris dilator muscle. This is unsurprising, but a more precise understanding of the temporal dynamics of the PRP response would be of interest for future studies.

Referee #2 comments:

I appreciate the authors addition of a controlled breathing experiment based on my question. I also appreciate the citations connecting controlled breathing exercises to the LC and to attention.

We are happy to hear that the changes made are appreciated.

Referee #3 comments:

The data analysis utilises advanced statistical methods well-suited to the dataset. However, the analysis would benefit from enhanced clarity, more detailed explanations, and explicit procedural steps to improve reproducibility and overall scientific rigour.

While the authors employed permutation testing as a post-hoc statistical analysis to identify breathing bins in which pupil size significantly deviated from random distributions, they did not apply corrections for multiple comparisons. Conducting 18 individual tests without adjustment increases the likelihood of a false positive result to approximately 60%. To address this, correction methods such as the Bonferroni adjustment, Benjamini-Hochberg procedure, or permutation-based family-wise error rate corrections should be considered and explicitly reported.

Thank you for this suggestion. Based on the feedback from Referees 1 and 3 we changed our permutation testing approach and included a permutation-based family-wise error rate correction (see the updated methods section). This resulted in more conservative p-values and some breathing bins no longer display normalized pupil sizes significantly different from zero. We have adjusted the figures and p-values accordingly, but the changes have not affected any of our conclusions.

The use of repeated measures ANOVAs is appropriate; however, the manuscript does not

specify whether key assumptions, such as sphericity, were tested. If sphericity was violated, corrections such as the Greenhouse-Geisser adjustment should have been applied, and these adjustments need to be reported. Furthermore, the assumption of independence in ANOVA requires careful attention to avoid pseudo-replication, where each row of data should represent a distinct participant. If nested or repeated measures data structures are present, linear mixed models incorporating participant IDs as random effects may be a more suitable analytical approach. The authors are encouraged to clarify how they have addressed potential pseudo-replication issues and ensured the independence of their data within the ANOVA framework.

We have now checked for sphericity violations and applied Greenhouse-Geisser adjustments when necessary. This is led to an adjustment of p-values and degrees of freedom, but did not affect any of our conclusions. We explain the procedure in the methods section in the following manner:

For each ANOVA we performed a Mauchly's test to check whether the assumption of sphericity was violated for the main effects or their interaction. When the assumption of sphericity was violated we applied a Greenhouse-Geisser correction. In the one case where sphericity could not be assessed due to a singular sum of squares and products matrix, we performed the nonparametric Friedman test instead.

Regarding the potential for pseudo replication, the data tables underlying our repeated measures ANOVA are organized in the way that each row holds the data for a distinct participant. The inherent correlation of repeated measures on the same subject are accounted for by the repeated measures ANOVA. The degrees of freedom in the ANOVA table also match up with the expected number based on our experimental design. Therefore, we do not believe our ANOVA analyses to suffer from pseudo replication.

Dear Dr Schaefer,

Re: JP-RP-2025-287205R2 "The Pupillary Respiratory-Phase Response: Pupil size is smallest around inhalation onset and largest during exhalation" by Martin Schaefer, Sebastiaan Mathot, Mikael Lundqvist, Johan N Lundstrom, and Artin Arshamian

We are pleased to tell you that your paper has been accepted for publication in The Journal of Physiology.

Yours sincerely,

Harold Schultz
Senior Editor
The Journal of Physiology

If you would like to receive our 'Research Roundup', a monthly newsletter highlighting the cutting-edge research published in The Physiological Society's family of journals (The Journal of Physiology, Experimental Physiology, Physiological Reports, The Journal of Nutritional Physiology and The Journal of Precision Medicine: Health and Disease), please click this link, fill in your name and email address and select 'Research Roundup':
<https://www.physoc.org/journals-and-media/membernews>

- You can help your research get the attention it deserves! Check out Wiley's free Promotion Guide for best-practice recommendations for promoting your work at: www.wileyauthors.com/eoo/guide. You can learn more about Wiley Editing Services which offers professional video, design, and writing services to create shareable video abstracts, infographics, conference posters, lay summaries, and research news stories for your research at: www.wileyauthors.com/eoo/promotion.

Reviewing Editor's comments to the authors:

I thank the authors for addressing all concerns and making the necessary changes in the text. The referees and I have no further comments. Congratulations on this publication.

Senior Editor's comments:

The editors wish to thank the authors for their thorough and thoughtful final adjustments to the manuscript. The article is now accepted for publication. Congratulations for an interesting and insightful study. Please consider the Journal of Physiology for your future studies.

Referee #1:

The authors have satisfactorily addressed all my concerns. The additional text and statistical analysis make the manuscript clearer and strengthen the rigor, interpretation, and conclusions of the manuscript.

Referee #2:

I have no further concerns or questions regarding the manuscript.

END OF COMMENTS